# Striatal dopamine D2/D3 receptor regulation of human reward processing and behaviour

Martin Osugo [1,2,3] ✉, Matthew B. Wall[4,5], Pierluigi Selvaggi[6,7], Uzma Zahid[1,8], Valeria Finelli [1], George E. Chapman[1,2,9,10], Thomas Whitehurst[2,11], Ellis Chika Onwordi[1,2,11,12], Ben Statton [2,5], Robert A. McCutcheon[1,2,13,14], Robin M. Murray [1], Tiago Reis Marques [1,2], Mitul A. Mehta [6] & Oliver D. Howes [1,2,3] ✉

Signalling at dopamine D2/D3 receptors is thought to underlie motivated behaviour, pleasure experiences and emotional expression based on animal studies, but it is unclear if this is the case in humans or how this relates to neural processing of reward stimuli. Using a randomised, double-blind, placebo-controlled, crossover neuroimaging study, we show in healthy humans that sustained dopamine D2/D3 receptor antagonism for 7 days results in negative symptoms (impairments in motivated behaviour, hedonic experience, verbal and emotional expression) and that this is related to blunted striatal response to reward stimuli. In contrast, 7 days of partial D2/D3 agonism does not disrupt reward signalling, motivated behaviour or hedonic experience. Both D2/D3 antagonism and partial agonism induce motor impairments, which are not related to striatal reward response. These findings identify a central role for D2/D3 signalling and reward processing in the mechanism underlying motivated behaviour and emotional responses in humans, with implications for understanding neuropsychiatric disorders such as schizophrenia and Parkinson's disease.

Identifying and responding appropriately to stimuli that signal a reward is fundamental to survival[1]. Dysfunction of the reward system is hypothesised to lead to impairments in motivated behaviour, hedonic experience, and emotional responses (collectively termed negative symptoms) in several disorders, including schizophrenia, substance use disorders, and Parkinson's disease[2-4]. Collectively, these disorders affect ~200 million people worldwide[5-7], highlighting the importance of understanding the neural mechanisms regulating reward processing and how these lead to negative symptoms.

Signalling at striatal dopamine D2/D3 receptors is proposed to play a central role in the neural processing of reward stimuli[1]. Supporting this, preclinical studies show that antagonism of D2/D3 receptors impairs reward-related learning and motivation to obtain rewards, and similar findings are seen in D2 receptor knockout mice[8-10]. The central role of the striatum is underlined by studies showing that striatal specific D2/D3 manipulations, including intrastriatal injections of D2/D3 receptor antagonists or genetic alterations of striatal D2 receptor expression, alter reward-related learning[11-13].

[1]Department of Psychosis Studies, Institute of Psychiatry, Psychology & Neuroscience, King's College London, London, UK. [2]MRC Laboratory of Medical Sciences, Imperial College London, London, UK. [3]South London and Maudsley NHS Foundation Trust, London, UK. [4]Perceptive, London, UK. [5]Faculty of Medicine, Imperial College London, London, UK. [6]Department of Neuroimaging, Institute of Psychiatry, Psychology and Neuroscience, King's College London, London, UK. [7]Department of Translational Biomedicine and Neuroscience, University of Bari "Aldo Moro", Bari, Italy. [8]Department of Psychology, Institute of Psychiatry, Psychology & Neuroscience, King's College London, London, UK. [9]Division of Psychiatry, Faculty of Brain Sciences, University College London, London, UK. [10]North London NHS Foundation Trust, London, UK. [11]East London NHS Foundation Trust, London, UK. [12]Centre for Psychiatry and Mental Health, Wolfson Institute of Population Health, Queen Mary University of London, London, UK. [13]Department of Psychiatry, University of Oxford, Oxford, UK. [14]Oxford Health NHS Foundation Trust, Oxford, UK. ✉e-mail: martin.osugo@kcl.ac.uk; oliver.howes@kcl.ac.uk

In humans, positron emission tomography imaging studies show that striatal D2/D3 receptor function is associated with neural response to reward and reward-related learning[14,15]. Further observational evidence in humans comes from associations between striatal dopaminergic function and striatal response to reward stimuli in neuropsychiatric disorders[16]. However, inferences on causation are challenging in clinical samples due to baseline abnormalities in dopamine function and because patients are often treated with dopamine-modulating drugs. To test the role of D2/D3 signalling experimentally in humans, several studies have examined neural responses to reward stimuli in healthy humans after pharmacological challenges. These studies show that dopamine D2/D3 receptor antagonism reduces striatal activation upon the receipt of reward stimuli and during reward-related learning[1,17-20]. However, it remains unknown how this relates to alterations in motivated behaviour and hedonic and emotional responses in humans[1]. Moreover, as all experimentally controlled studies to date used single doses of D2/D3 antagonists, it remains unclear if effects persist or if the system adapts when D2/D3 antagonism is sustained. This is an important question as tens of million people worldwide take long-term antipsychotic drugs each year[21,22].

We, therefore, aimed to establish the effect of sustained D2/D3 receptor antagonism on the neural processing of reward-related stimuli and behaviour in healthy humans using a randomised, double-blind, placebo-controlled, crossover design (Fig. 1). Based on the single-dose studies[1], we hypothesised that, compared to placebo, the D2/D3 receptor antagonist amisulpride would reduce reward-related brain activation in the striatum, and that this would be correlated with impairments in motivated behaviour and hedonic experience induced by amisulpride. To further probe the role of D2/D3 receptors in mediating striatal responses, we conducted a further study using the D2/D3

receptor partial agonist aripiprazole, as in animal studies D2/D3 partial agonists have been shown to suppress phasic dopamine signalling more than tonic signalling, whereas antagonists have been shown to suppress both equally[23,24]. This work advances understanding of the regulation of human behaviour, experience and motor function by striatal D2/D3 receptors.

## Results

Two independent healthy volunteer cohorts received either amisulpride and placebo, or aripiprazole and placebo for seven days each in a randomised, double-blind, placebo-controlled, crossover design (Fig. 1). The amisulpride and placebo crossover study (arm 1) and aripiprazole and placebo crossover study (arm 2) were conducted sequentially at the same site, with no differences between the arms in terms of key study personnel, setting, recruitment strategy, inclusion/exclusion criteria, study design, data acquisition or data management/analysis (see 'Methods' and Supplementary methods for further details). Seventy-six healthy volunteers with no history of neuropsychiatric disorder were recruited and fifty completed the experiments (see Supplementary for details). There were no significant differences between the amisulpride ($n = 25$) and aripiprazole ($n = 25$) samples on any demographic variables or on the proportion receiving placebo first. The samples are further described in Table S1.

Outcomes were assessed at baseline and following administration of active drug and placebo. The Brief Negative Symptom Scale (BNSS) was used to assess motivated behaviour, hedonic responses, and emotional expressivity, impairments of which are termed negative symptoms. Negative symptoms are common, with a prevalence of 60% in schizophrenia, >40% in Parkinson's disease and >50% in Alzheimer's disease[25]. The BNSS is sensitive to change, can detect individual differences in negative symptoms in the healthy population and has been

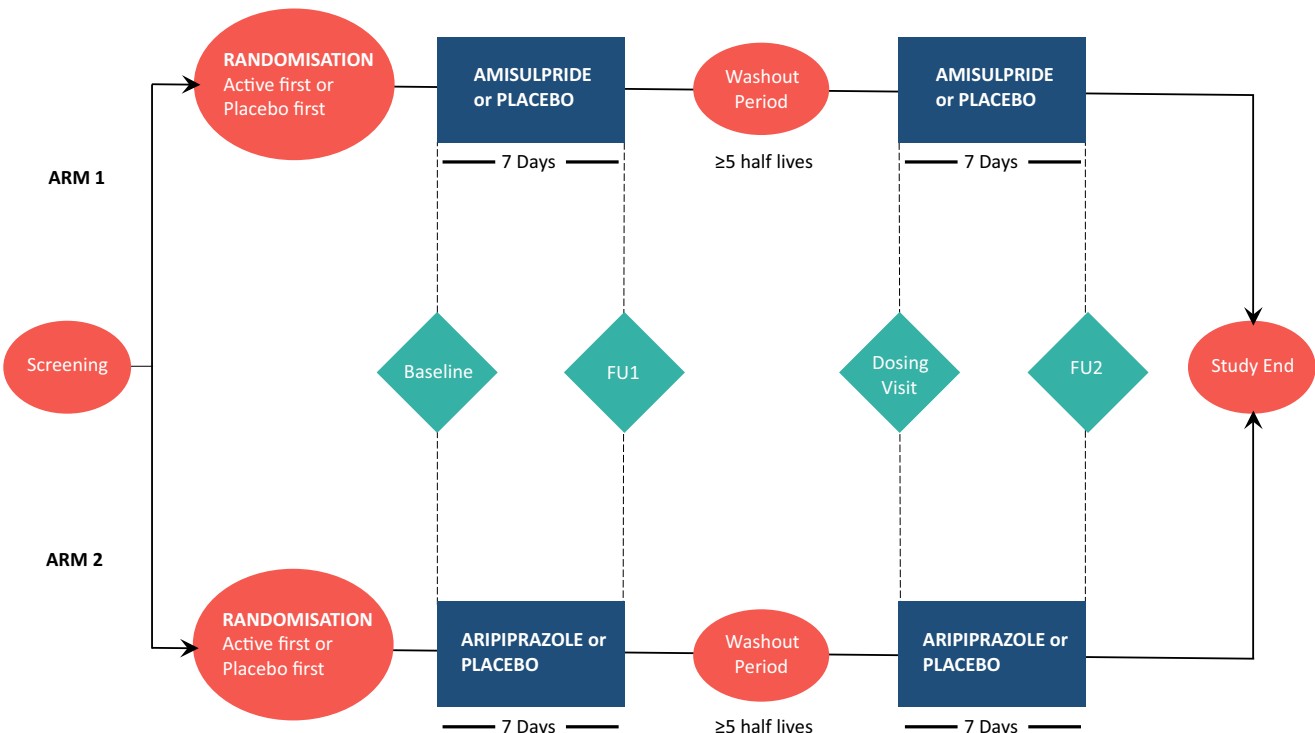

**Fig. 1 | Study design.** The order of study interventions and measures for the two arms of the study is shown. The arms were conducted sequentially, with arm 1 completed prior to arm 2 commencing. The order of treatments within each arm (active drug or placebo first) was randomised and counter-balanced to ensure approximately equal numbers of subjects receiving drug or placebo first. Subjects and investigators were blind to the treatment allocation. The washout period was a minimum of 10 days for amisulpride, and a minimum of 28 days for aripiprazole. The monetary incentive delay (MID) task, Brief Negative Symptom Scale (BNSS), Simpson Angus Scale (SAS) and Barnes Akathisia Rating Scale (BARS) were performed at baseline, follow up (FU) 1 and FU 2 visits, to measure the neural response during reward processing and behavioural, hedonic, and motor effects.

related to deficits in reward processing in several studies, making it suitable to measure clinically relevant changes in hedonic responses and related behaviour and emotional expression[26–28]. We assessed motor function using the Simpson Angus Scale (SAS) to assess rigidity and bradykinesia (parkinsonism), and the Barnes Akathisia Rating Scale (BARS) to assess motor restlessness.

## Behavioural and motor effects of the D2/D3 receptor antagonist amisulpride

Compared with placebo, dopamine D2/D3 antagonism with amisulpride increased total scores on the BNSS ($b = 4.62$, $t(49) = 2.51$, 95% CI = 0.92:8.32, $p = 0.015$). It also resulted in motor impairments of akathisia (BARS: $b = 1.48$, $t(50) = 4.53$, 95% CI = 0.83:2.14, $p = 3.63 \times 10^{-5}$) and parkinsonism (SAS: $b = 1.03$, $t(50) = 3.12$, 95% CI = 0.37:1.69, $p = 0.0030$) (Fig. 2A–C). Exploratory analysis of the BNSS two-factor model indicated that amisulpride led to expressive deficits compared to placebo ($p = 0.0057$), but did not affect the motivation and pleasure subscale ($p = 0.13$). Exploratory analysis of the BNSS five-factor model indicated that amisulpride increased ratings of blunted affect ($p = 0.0073$) and alogia ($p = 0.011$) compared to placebo, but not ratings of anhedonia ($p = 0.35$), asociality ($p = 0.10$), or avolition ($p = 0.18$). For full results see Supplementary Tables S2–S6.

## Effects of D2/D3 receptor antagonist amisulpride on reward response

We measured the neural response to reward stimuli using the monetary incentive delay (MID) task, an event-related functional magnetic resonance imaging (fMRI) task which probes brain activation upon expectation and receipt of monetary reward (for further details see methods and supplements). At baseline, we detected the expected brain activations in the bilateral striatum and related cortical reward structures (insula, prefrontal cortex, anterior cingulate) during reward anticipation and reward outcome (Supplementary Figs. S1 and S2). We did not find any effect of amisulpride or of practice on task performance in terms of overall accuracy, overall reaction time, reward trial accuracy or reward trial reaction time (full results in supplements).

During reward anticipation, there were no significant differences between the amisulpride and placebo conditions in pre-specified striatal region of interest (ROI) analysis (Supplementary Table S7). During reward outcome, in our pre-specified striatal ROIs, we found a medium-to-large effect size reduction in blood oxygen level-dependent (BOLD) signal in the bilateral caudate on amisulpride compared to placebo (paired t-test, $t(20) = -3.17$, FDR corrected $p$-value = 0.014, Cohen's $d = -0.69$; Fig. 2D). There were no significant differences in the putamen or nucleus accumbens (Supplementary Table S7). Exploratory whole brain analyses found significant BOLD reductions during reward outcome on amisulpride compared to placebo in multiple regions, with peaks in the orbito-frontal cortex, right caudate and inferior frontal gyrus, and extension across anterior cingulate gyrus, paracingulate gyrus, frontal pole, superior frontal gyrus, insula, opercular cortex, left caudate and the putamen (Table 1 and Fig. 2G and S3). These are regions that prior meta-analyses suggest are central to reward processing[29]. There were no significant differences between amisulpride and placebo in whole brain analysis during reward anticipation.

To address potential confounding effects on the BOLD signal from changes in cerebral blood flow following amisulpride, we conducted a further analysis comparing the amisulpride-placebo reward outcome BOLD signal in the caudate after covarying for regional cerebral blood flow (rCBF) in the same region in the two treatment conditions[17]. The observed effect of amisulpride in blunting caudate BOLD response remained significant after covarying for caudate rCBF, ($b = 158.2$, $t(34) = 3.63$, 95% CI = −65.92:−246.9, $p = 0.00093$), with a larger effect size despite a 25% reduction in sample size (due to unusable or missing rCBF data) compared to the linear mixed model which

did not include caudate rCBF ($b = 122.9$, $t(46) = 2.75$, $p = 0.0085$). There was no relationship between caudate rCBF and caudate reward outcome BOLD signal in the linear mixed model ($p = 0.54$).

Having shown that amisulpride reduced caudate activation to reward stimuli, we tested the relationship between caudate activation and plasma amisulpride levels. We found that a greater reduction in caudate activation from baseline was associated with higher amisulpride levels ($n = 22$, $r = -0.44$, FDR corrected $p$-value = 0.041; Fig. 2E), providing further evidence that the caudate effect was due to amisulpride exposure. The reduction in BOLD signal in the caudate during reward outcome between the baseline and amisulpride scans was also correlated with the increase in BNSS total scores ($n = 21$, Spearman's rho = −0.49, FDR corrected $p$-value = 0.041; Fig. 2F). This relationship was specific to amisulpride, as there was no relationship between the change in the two variables between the baseline and placebo assessments ($n = 22$, Spearman's rho = −0.002, $p = 0.99$), and comparing the two correlation coefficients showed that the baseline–placebo correlation was significantly different from the baseline–amisulpride correlation ($n = 18$, ZPF = −2.05, $p = 0.041$)[30].

We then went on to conduct further exploratory analyses of potential mechanisms associated with the symptom changes demonstrated. We found that the increase in total negative symptoms on the BNSS between the baseline and amisulpride assessments was highly correlated with the change in Parkinsonian symptoms on the SAS over the same time period ($n = 27$, rho = 0.62, $p = <0.001$), but was not related to the change in symptoms of akathisia on the BARS ($n = 27$, rho = 0.23, $p = 0.25$). Amisulpride-induced parkinsonian symptoms were related to both the expressive ($n = 27$, rho=0.55, $p = 0.003$) and motivation/pleasure domains of the BNSS ($n = 27$, rho = 0.44, $p = 0.022$). The effect size of the relationship with the expressive domain was greater numerically, but did not differ statistically from the correlation with motivational deficits ($n = 27$, William's $T = 0.68$, $p = 0.50$)[31].

We found no relationship in exploratory correlations between the reduction in caudate reward signal and the increase in motor symptoms between the baseline and amisulpride assessments (SAS: $n = 21$, rho = −0.027, $p = 0.91$; BARS: $n = 21$, rho = 0.34, $p = 0.13$). Exploratory correlations between the change in the BNSS factors showed that the change in caudate reward signal was not correlated with the change in the BNSS expressive factor ($n = 21$, rho = −0.41, $p = 0.064$) or the BNSS motivation and pleasure factors of the two-factor model ($n = 21$, rho = −0.40, $p = 0.076$), but was related to the change in the BNSS avolition factor ($n = 21$, rho = −0.46, $p = 0.037$) of the five-factor model (for full results see Table S8).

Having shown that the D2/D3 antagonist amisulpride reduced the striatal response to reward stimuli, and increased negative symptoms and motor impairments, we sought to determine if a partial D2/D3 agonist had similar effects, to test the effects of maintaining tonic D2/D3 signalling whilst supressing phasic D2/D3 signalling.

## Behavioural and motor effects of the D2/D3 receptor partial agonist aripiprazole

Compared with placebo, the dopamine D2/D3 partial agonist aripiprazole did not lead to negative symptoms overall (BNSS: $b = 1.24$, $t(48) = 1.69$, 95% CI = −0.24:2.72, $p = 0.10$), or on the expressive ($p = 0.18$), motivation/pleasure ($p = 0.14$), anhedonia ($p = 0.10$), asociality ($p = 0.99$), avolition ($p = 0.23$), blunted affect ($p = 0.30$) or alogia ($p = 0.11$) subscales of the BNSS. However, it caused akathisia (BARS: b = 1.04, $t(49) = 3.31$, 95% CI = 0.41:1.67, $p = 0.0017$) and parkinsonism (SAS: $b = 1.19$, $t(48) = 3.61$, 95% CI = 0.53:1.85, $p = 0.00073$) (Fig. 3A–C). For full results, see Supplementary Tables S9–S13.

## Effects of D2/D3 receptor partial agonist aripiprazole on reward response

We did not find any effect of aripiprazole or of practice on task performance in terms of overall accuracy, overall reaction time,

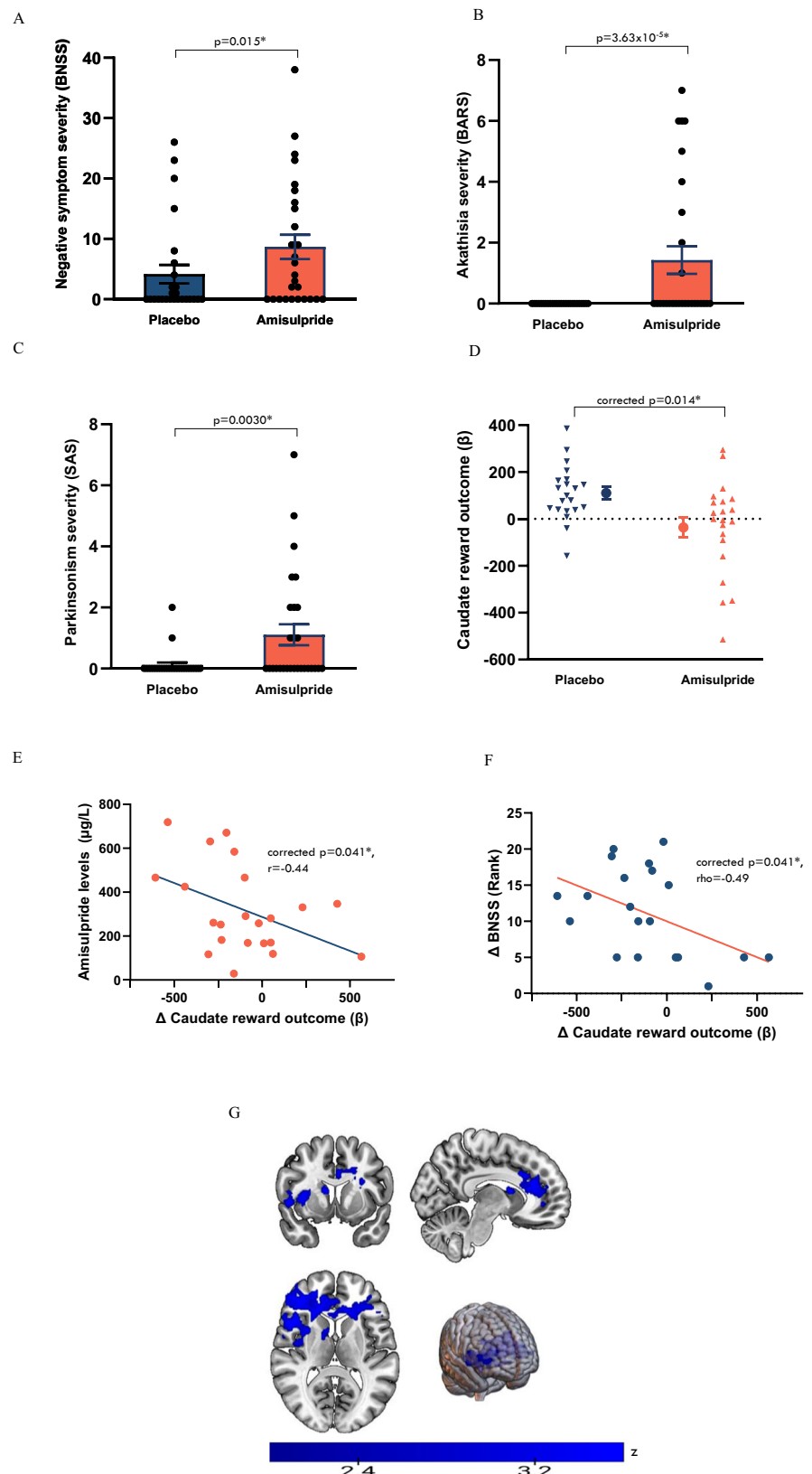

reward trial accuracy or reward trial reaction time (full results in supplements). There was no effect of aripiprazole on the neural response to reward anticipation or reward outcome, in the pre-specified striatal ROIs or in whole brain analysis (Fig. 3D and Supplementary Table S14).

## Comparison between D2/D3 antagonism and D2/D3 partial agonism

We did not find evidence that, relative to placebo, amisulpride induced greater overall negative symptoms than aripiprazole (BNSS: $t(54) = -1.71$, $p = 0.094$). We also found no difference between

**Fig. 2 | Effects of the D2/D3 antagonist amisulpride on human-motivated behaviour, hedonic and emotional responses, motor function and reward processing. A** Effect of amisulpride vs placebo on negative symptoms (Brief Negative Symptom Scale, individual values with mean ± SE). $n = 29$, linear mixed-effects model: $b = 4.62$, $t(49) = 2.51$, 95% CI = 0.92:8.32, two-sided p = 0.015. **B** Effect of amisulpride vs placebo on akathisia (Barnes Akathisia Rating Scale, individual values with mean ± SE). $n = 29$, linear mixed effects model: $b = 1.48$, $t(50) = 4.53$, 95% CI = 0.83:2.14, two-sided $p = 3.63 \times 10^{-5}$. **C** Effect of amisulpride vs placebo on parkinsonism (Simpson Angus Scale, individual values with mean ± SE). $n = 29$, linear mixed effects model: $b = 1.03$, $t(50) = 3.12$, 95% CI = 0.37:1.69, two-sided $p = 0.0030$. **D** Effect of amisulpride vs placebo on caudate activation during reward outcome in monetary incentive delay (MID) task (individual values with mean ± SE of beta values). $n = 21$, paired sample $t$-test: mean difference = −146.2, $t(20) = 3.17$, 95% CI = −50.0:−242.4, FDR corrected two-sided $p = 0.014$. **E** Relationship between change in caudate BOLD signal during reward outcome on amisulpride and plasma amisulpride levels (µg/L). $n = 22$, Pearson's $r = −0.44$, FDR corrected $p$-value = 0.041. **F** Relationship between reduction in caudate reward outcome activation and ranked increase in negative symptoms on amisulpride (BNSS). $n = 21$, Spearman's rho = −0.49, FDR corrected $p$-value = 0.041. **G** Whole-brain statistical maps showing areas of reduced BOLD signal on amisulpride vs placebo during reward outcome. $n = 21$. Colour bar indicates z-statistic. Source data are provided as a Source data file.

**Table 1 | Peak of clusters showing significant reduction in brain activation on amisulpride compared to placebo during reward outcome in whole brain analysis (FSL randomise: two-sided paired t-test comparing amisulpride to placebo, with 5000 permutations and FWE correction for multiple comparisons using threshold-free cluster enhancement at p < 0.05)**

| Cluster peak | Number of voxels | Cluster peak co-ordinates (MNI) | Cluster p-value (FWE corrected) | Z-MAX |
|---|---|---|---|---|
| Frontal orbital cortex | 6843 | 46, 28, −8 | 0.018 | 3.54 |
| Right caudate | 98 | 8,8, 10 | 0.045 | 3.54 |
| Inferior frontal gyrus, pars triangularis | 21 | −54,26, 6 | 0.046 | 3.54 |

amisulpride–placebo and aripiprazole–placebo on the expressive ($t(54) = −1.69$, $p = 0.097$) and motivation/pleasure subscales of the BNSS ($t(54) = −1.00$, $p = 0.32$). However, we found exploratory evidence that amisulpride led to greater blunted affect relative to placebo than aripiprazole ($t(54) = −2.09$, $p = 0.041$), with no difference between the effects of the drugs relative to placebo on the other BNSS factors or on motor symptoms (SAS: $t(54) = 0.33$, $p = 0.74$; BARS: t(54) = 0.97, $p = 0.33$) (Fig. 4A–C).

During reward outcome, in pre-specified striatal ROI analysis, we found a significantly greater reduction in BOLD activation between amisulpride–placebo compared to aripiprazole–placebo in the caudate, with a large effect size (t(41) = 2.75, amisulpride mean = −146.19 ± 211.33, aripiprazole mean = 26.84 ± 201.99, FDR corrected $p$-value = 0.027, Cohen's $d = 0.84$; Fig. 4D). There were no significant differences between the drugs in the putamen or nucleus accumbens during reward outcome, or in any striatal ROI during reward anticipation (supplementary Table S15). In exploratory whole-brain analysis, we found significantly greater reductions in neural activation during reward outcome on amisulpride relative to placebo than on aripiprazole relative to placebo, with the largest peaks in the opercular cortex and caudate (Table 2 and Fig. 4E). There was an extension of these clusters across the reward network into the anterior cingulate gyrus, paracingulate gyrus, frontal pole and the middle and inferior frontal gyri (Supplementary Fig. S4).

To test whether these differences could be attributed to baseline differences between subjects in the two arms in the outcome variables, we conducted independent sample t-tests comparing baseline MID task performance measures, baseline striatal reward signal during reward anticipation and reward outcome, baseline motor symptoms and baseline negative symptoms between the two arms. There were no significant differences between subjects who were subsequently enroled in arm 1 and subjects who were subsequently enroled in arm 2 on any of these measures (all FDR corrected $p$-values > 0.05, Supplementary Table S16). We conducted a similar analysis to ensure that differences in placebo response did not contribute to the observed differences. We again found no significant differences between subjects in the amisulpride arm and subjects in the aripiprazole arm following their respective placebo conditions, or in the change from baseline assessment to the placebo assessment, in terms of task performance, striatal reward signal, motor function or negative symptoms (all FDR corrected $p$-values > 0.05, Table S16).

## Discussion

Although it has long been thought that striatal signalling to reward stimuli plays a role in negative symptoms (impairments in motivated behaviour, hedonic responses and emotional expressivity), there has been scant causal evidence for this link in humans. Here we show that sustained dopamine D2/D3 receptor antagonism results in blunted striatal responses to rewards in healthy humans. We also show that, in the absence of impairments in task performance, that more blunted striatal response to reward is associated with greater induction of motivational and expressive deficits, but is not related to motor impairments. In contrast, we show that D2/D3 receptor partial agonism does not disrupt reward signalling, motivated behaviour and pleasure, but does induce similar motor impairments to D2/D3 antagonism.

The doses of amisulpride and aripiprazole we used have been shown to induce substantial (>60%) striatal D2/D3 receptor occupancy in humans[32]. Our work demonstrates that D2/D3 receptors have a key role in regulating striatal responses to reward stimuli and to reward related behaviour. An influential model proposes that the striatum regulates brain circuits and behaviour, including reward and motor function[33], through two main output pathways: an excitatory direct pathway regulated by D1 receptors and an inhibitory indirect pathway regulated by D2/D3 receptors[34,35]. As dopamine is inhibitory at D2/D3 receptors, D2/D3 antagonism is proposed to result in increased inhibitory output from the indirect pathway, leading to decreased movement[34–36]. Our findings that the D2/D3 receptor antagonist amisulpride caused parkinsonian symptoms of rigidity and bradykinesia are consistent with this model. We further demonstrate that D2/D3 inhibition of the inhibitory pathway is an important part of the neural response to reward stimuli. However, we found that both D2/D3 antagonism and partial agonism led to akathisia, which is characterised by excessive movement. This may be explained by imbalance between activities of the direct and indirect pathways following activation of the indirect pathway, as excitatory D1 receptors in the direct pathway were not affected by the D2/D3 modulators administered in our study[37].

In contrast to D2/D3 antagonists, aripiprazole has about 25% of the intrinsic activity of dopamine at D2/D3 receptors[38]. Our finding that aripiprazole did not blunt striatal reward response or induce negative symptoms indicates that maintaining some tonic activation of D2/D3 receptors is sufficient to permit normal striatal reward signalling and behavioural responses. In contrast, our observation that partial agonism at this level is not sufficient to prevent motor impairments demonstrates the role of phasic D2/D3 signalling in the indirect

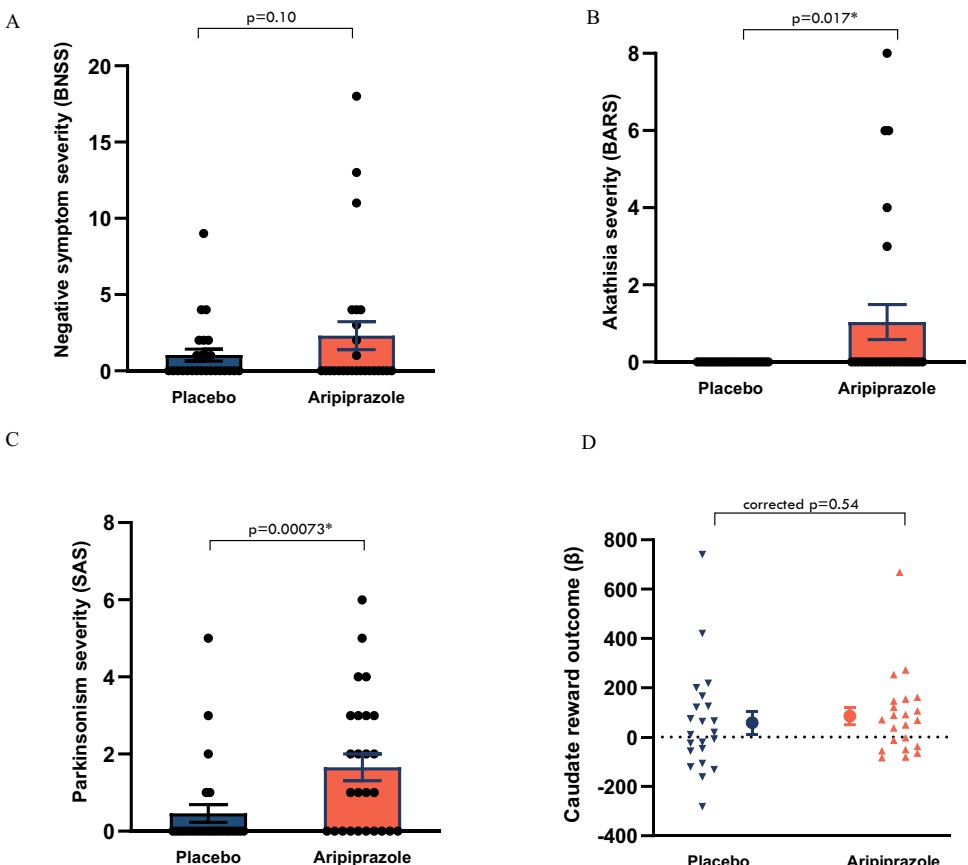

**Fig. 3 | Effects of the D2/D3 partial agonist aripiprazole on human-motivated behaviour, hedonic and emotional responses, motor function and reward processing. A** Effect of aripiprazole vs placebo on negative symptoms (Brief Negative Symptom Scale, individual values with mean ± SE). $n = 27$, linear mixed effects model: BNSS: $b = 1.24$, $t(48) = 1.69$, 95% CI = −0.24:2.72, two-sided $p = 0.10$. **B** Effect of aripiprazole vs placebo on akathisia (Barnes Akathisia Rating Scale, individual values with mean ± SE). $n = 27$, linear mixed effects model: $b = 1.04$, $t(49) = 3.31$, 95% CI = 0.41:1.67, two-sided $p = 0.0017$. **C** Effect of aripiprazole vs placebo on parkinsonism (Simpson Angus Scale, individual values with mean ± SE). $n = 27$, linear mixed effects model: $b = 1.19$, $t(48) = 3.61$, 95% CI = 0.53:1.85, two-sided $p = 0.00073$. **D** Effect of aripiprazole vs placebo on caudate activation during reward outcome in monetary incentive delay (MID) task (individual values with mean ± SE of beta values). $n = 22$, paired sample $t$-test: mean difference = 26.8, $t(21) = 0.62$, 95% CI = −116.4:62.7, FDR corrected two-sided $p = 0.54$. Source data are provided as a Source data file.

pathway's inhibition of motor function, and provides evidence in humans that reward and motor function are mediated by different striatal pathways which are differentially sensitive to dopamine.

Several preclinical studies and observational studies in healthy humans show that dopamine signalling in the dorsal striatum is critical for reinforcement learning[12,39–46]. Single-dose studies in humans support this, consistently demonstrating reduced caudate and ventral striatal activation during reward-related learning following D2/D3 antagonism[20,47–49]. Our findings of a reduction in reward signal in the caudate following sustained D2/D3 antagonism, which correlated with the induction of negative symptoms, and the observed differences between the effects of D2/D3 antagonism and partial agonism on reward signalling in the caudate are consistent with this interpretation. However, the acute effects of D2/D3 antagonism on striatal activation during reward receipt in humans are mixed. Three studies found no effect of D2/D3 antagonism on striatal activity during reward receipt[17,18,50], whereas one showed reduced striatal activity[19]. These inconsistencies may be partly related to the acute nature of the drug challenges, as single doses of D2/D3 antagonists may not result in the same exposure as multiple dosing[51]. Moreover, emerging evidence suggests that antipsychotic accumulation in brain tissue following repeated dosing contributes to their pharmacological effects[52,53].

These inconsistencies may also be related to differences in the reward paradigms and incomplete understanding of reward

processing. Although it has been hypothesised that dopaminergic signalling is more related to reward anticipation than reward outcome, the validity of these theorised phases of reward processing is unclear, as are the molecular mechanisms underlying them[1]. Schizophrenia has been associated with abnormalities in both reward anticipation and reward outcome[54]. At the time of study conception, single-dose studies in healthy humans were equivocal, with one significant double-blind study showing reduced striatal reward response following acute D2/D3 antagonism for reward anticipation and reward outcome respectively (a further study showing reduced striatal response during reward anticipation has since been published[17–19]). We, therefore, lacked a clear hypothesis as to whether sustained D2/D3 modulation would preferentially affect reward anticipation or reward outcome, and chose to investigate effects on both phases, as was intended in the design of the MID task[55]. However, we used a 50% win rate in contrast to the 66% win rate in the original version of the MID, as a lower number of events per condition can lead to lower reliability[56]. The 50% win rate used in our study has been extensively used in prior research, and studies using it have been shown to activate similar brain regions to versions of the MID with a 66% win rate[57–59]. Nevertheless, a systematic investigation of the effect of hit rate on signal magnitude during the reward anticipation and reward outcome phases of the MID has not been conducted. We detected more extensive striatal activation at baseline

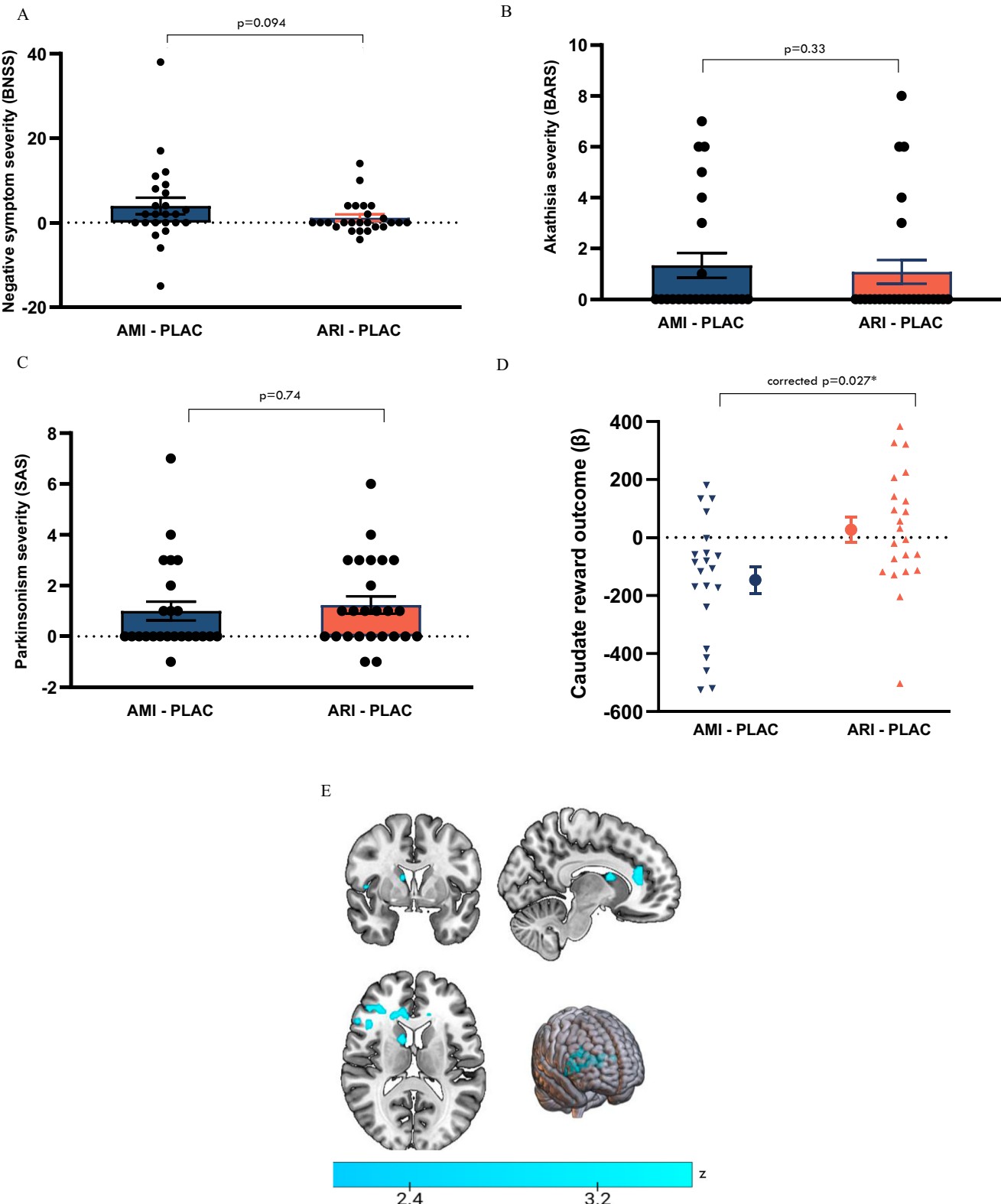

**Fig. 4 | Comparison between the effects of the D2/D3 antagonist amisulpride and the D2/D3 partial agonist aripiprazole on human-motivated behaviour, hedonic and emotional responses, motor function and reward processing.**
**A** Comparison of amisulpride–placebo (AMI–PLAC) differences to aripiprazole–placebo (ARI–PLAC) differences on negative symptoms (Brief Negative Symptom Scale, individual values with mean ± SE). $n = 56$, Student's $t$-test: $t(54) = -1.71$, two-sided $p = 0.094$. **B** Comparison of AMI–PLAC differences to ARI–PLAC differences on akathisia (Barnes Akathisia Rating Scale, individual values with mean ± SE). $n = 56$, Student's $t$-test: $t(54) = 0.97$, two-sided $p = 0.33$.
**C** Comparison of AMI–PLAC differences to ARI–PLAC differences on Parkinsonism

(Simpson Angus Scale, individual value with mean ± SE). $n = 56$, Student's $t$-test: $t(54) = 0.33$, two-sided $p = 0.74$. **D** Comparison of AMI–PLAC differences to ARI–PLAC differences on reward outcome activation in caudate from monetary incentive delay (MID) task (individual values with mean ± SE of beta values). $n = 43$, independent sample $t$-test: mean difference = =173.0, $t(41) = -2.75$, 95% CI: $-300.3$:$-45.7$, FDR corrected two-sided $p$-value = 0.027, Cohen's $d = 0.84$.
**E** Statistical map for brain regions where the reduction in neural response during reward outcome on AMI is greater than that on ARI, relative to placebo. $n = 43$. Colour bar indicates z-statistic. Source data are provided as a Source data file.

**Table 2 | Peak of clusters showing significantly greater reduction in neural response on amisulpride relative to placebo compared to aripiprazole relative to placebo during reward outcome in whole brain analysis (FSL randomise: two-sided independent sample t-tests comparing amisulpride–placebo to aripiprazole–placebo, with 5000 permutations and FWE correction for multiple comparisons using threshold-free cluster enhancement at $p < 0.05$)**

| Cluster peak | Number of voxels | Cluster peak co-ordinates (MNI) | Cluster p-value (FWE corrected) | Z-MAX |
|---|---|---|---|---|
| Frontal operculum cortex | 928 | 44, 18, 8 | 0.040 | 3.54 |
| Right caudate | 59 | 10, 2, 14 | 0.044 | 3.54 |
| Central opercular cortex | 26 | 46, -8, 6 | 0.048 | 3.09 |

during the reward outcome phase compared to the reward anticipation phase, which may have increased our ability to find drug effects during the reward outcome phase. This may be attributable to the relatively low magnitude reward utilised in our study (£0.30), as some evidence suggests that striatal response in the anticipation phase is more sensitive to reward magnitude than the reward outcome phase[57,60]. Overall, it is challenging to separate reward anticipation, reward outcome and reward learning in fMRI tasks, including the MID[1]. Future research should aim to test the effects of sustained D2/D3 modulation on reward processing using tasks which more clearly distinguish the putative phases of reward processing.

Our randomised, double-blind, placebo-controlled study design is well suited for making causative inferences, particularly as we controlled for important confounders including psychoactive drug use and drug-induced changes in regional blood flow[17]. We also did not find altered task performance following amisulpride or aripiprazole, as expected and in line with a previous study using the MID following a single dose of risperidone in healthy humans[17]. This further strengthens our conclusion that the reduced BOLD activation was due to reduced striatal activation to reward, and not to the confounder of impaired performance, which is also associated with alterations in BOLD amplitude[61]. Our findings therefore have implications for understanding neuropsychiatric disorders associated with striatal dopaminergic dysfunction such as Huntington's disease, Parkinson's disease and schizophrenia[62], by indicating the central role of D2/D3 receptors in mediating the striatal response to reward stimuli. They also have implications for the tens of millions of people who take dopamine D2/D3 receptor antagonist treatments each year, as they show that sustained D2/D3 antagonism disrupts reward signalling. Observational evidence shows links between dorsal striatal reward hypofunction and negative symptoms in antipsychotic-treated schizophrenia patients[63], and whilst impaired motivation, anhedonia and other negative symptoms are evident in people with schizophrenia who are not taking antipsychotic drugs, our findings indicate that treatment with D2/D3 antagonists may exacerbate them by impairing reward signalling. Considering that negative symptoms are common, highly impairing, and often intractable[64], and that dopamine antagonists are prescribed frequently and for long durations[65], this provides strong impetus towards the development of drugs with novel mechanisms to treat psychotic disorders[66].

In addition to providing causative evidence on the role of D2/D3 antagonists in the development of negative symptoms, our work advances understanding of the underlying mechanisms. We found no relationship between striatal reward deficits and motor impairments induced by amisulpride, but found that negative symptoms induced by amisulpride were related to both variables. Specifically, we found in exploratory analyses of the BNSS 5-factor model (which reflects current consensus negative symptom domains[67]) that blunted caudate D2/D3 striatal reward signalling may be particularly related to avolition, and that parkinsonian symptoms were related to both the expressive deficits and motivation/pleasure domains of the BNSS. Whilst the BNSS total score was our primary outcome, this provides preliminary causative evidence to support current hypotheses on the pathophysiology of avolition, drawn from studies conducted in

medicated schizophrenia patients which demonstrated relationships between blunted caudate response to reward and avolition, and studies in animals[63,68,69]. However, we note that these exploratory analyses were not corrected for multiple comparisons, highlighting the need for further investigations to confirm them. The relationship between parkinsonian symptoms and both motivation/pleasure and expressive domains is more difficult to interpret, particularly as aripiprazole caused parkinsonian symptoms to a similar degree as amisulpride but did not induce negative symptoms. These data suggest that there may be multiple mechanisms underlying antipsychotic-induced negative symptoms, which require further elucidation, for example with more sensitive computerised measures of expressive deficits and motor function.

Although we did not find any evidence of differences in baseline responses, placebo response, or in the change from baseline to placebo assessment between the amisulpride and aripiprazole arms, we acknowledge that this represents a potential confounder in interpreting differences in drug effects between arm 1 and arm 2. An alternative study design is a three-intervention crossover (the same subjects administered placebo, amisulpride and aripiprazole). However, order effects (including carry-over effects and practice effects) would be a greater issue than in the design we used because subjects would perform each outcome measure four times. In addition, such a design would be practically challenging, with each arm requiring the 28-day washout of aripiprazole in order to maintain blinding, in addition to the likelihood of a higher dropout rate due to subject burden and adverse events.

Pharmacokinetic or pharmacodynamic differences between aripiprazole and amisulpride may also account for the observed differences. Amisulpride is highly selective for dopamine D2/3 receptors and has very similar affinity to aripiprazole for D2/D3 receptors[70]. A potential consideration is that aripiprazole also has affinity for 5-HT1a, 5-HT2a, 5-HT2c, histamine-1 and alpha-1 receptors in addition to its partial agonism of dopamine D2/D3 receptors[71]. However, its affinity for these other receptors is 4–5-fold lower than its affinity for D2/D3 receptors, and receptor occupancy following repeated dosing is 30–70% lower, indicating that the effects we observe are more likely related to actions at D2/D3 receptors[71,72]. Amisulpride and aripiprazole act on D2/D3 receptors throughout the brain as well as in all striatal subdivisions; so it is possible that D2/D3 modulation elsewhere in the brain contributes to the effects observed in the caudate. This is challenging to investigate experimentally in humans, as striatal-specific manipulations of D2/D3 receptor function are not possible, however, D2/D3 receptor density is highest in the striatum and the effects of antipsychotics on brain haemodynamics have been shown to scale with the density of receptors[73]. Whilst the doses used in the study are considered the minimum clinically effective dose by the Maudsley Prescribing Guidelines, equivalent doses of antipsychotics are imprecise and 400 mg of amisulpride is proposed to be equivalent to 15 mg of aripiprazole by the same guidelines[74]. However, a recent meta-analysis found that aripiprazole at 10 mg/day was very close to the near-maximal response[75]. Moreover, 10 mg of aripiprazole actually corresponds to greater striatal D2/D3 occupancy (~80%) in comparison to 400 mg of amisulpride (~60%)[32], supporting our interpretation that

the drug differences observed are related to partial agonism by aripiprazole at D2/D3 receptors. We interpreted the differences between aripiprazole and amisulpride as relating to preferential inhibition of phasic signalling by the partial agonist, in comparison to inhibition of both tonic and phasic dopamine signalling equally by the antagonist. We however acknowledge that this distinction has only been demonstrated in animals, and that it is unclear whether this is the case in humans[23,24].

We also acknowledge that aripiprazole takes 14 days to reach steady state, whereas amisulpride reaches steady state in 3 days[76,77]. Although this is another important potential consideration, a PK/PD study of aripiprazole in healthy volunteers found that exposure (area under the curve over 24 h) following 8 days of aripiprazole at 10 mg daily was within 10% of exposure at 14 days[78]. In our study, antipsychotic levels were collected as close as possible to outcome data, with the maximum interval of a few hours making it unlikely that plasma level fluctuations influenced the results, considering the 94-h half-life of aripiprazole and its active metabolite. Additionally, the mean plasma aripiprazole concentration in our study corresponds to that associated with ~80% striatal D2/D3 occupancy[32]. For these reasons, we think it is unlikely that an inadequate dose or duration of aripiprazole treatment explains the drug differences, although future studies could test this further. Finally, all previous double-blind investigations of the effects of antipsychotics on reward function in healthy volunteers have used only single doses, whereas our study more closely mimics the clinical use of antipsychotics by investigating repeated administration.

Our finding that amisulpride induced negative symptoms may seem surprising, as there is some evidence that low dose (50–300 mg) amisulpride is effective as a treatment for negative symptoms in schizophrenia[79]. This has been hypothesised to relate to lower doses binding to presynaptic auto-receptors to enhance dopamine transmission, despite a lack of clear mechanistic evidence to support this[79]. Whether the effects of amisulpride and aripiprazole on negative symptoms and reward functioning are dose and duration dependent is a critical factor for future studies to investigate, in order to support the development of translational models to predict the inducement of negative symptoms by antipsychotics.

In conclusion, our study demonstrates that dopamine D2/D3 receptor signalling regulates human reward processing in the striatum and motivational and hedonic behaviour, and shows that reward and motor function in humans are mediated by different striatal pathways.

## Methods

### Ethical approval
This study was approved by the London – West London and GTAC NHS Research Ethics Committee (Ethics Committee Reference Number: 18/LO/1044). All subjects provided written, informed consent prior to participation.

### Participants
Healthy volunteers aged 18–65 years were recruited by public advertisement. Exclusion criteria were; history of psychiatric illness (including alcohol/substance dependence or abuse, other than caffeine/nicotine) as determined by self-report and the Mini-International Neuropsychiatric Interview; current use of any illicit substances as determined by urine drug of abuse testing and self-report; pregnancy as determined by urine pregnancy testing and self-report; self-report of a first degree relative with a psychotic disorder, current or significant previous use of psychotropic or dopamine modulating drugs, breastfeeding, or participation in a study of unlicensed medicines within the previous 30 days; self-report or clinical findings of significant CNS disorder (e.g. significant head trauma, epilepsy, etc.), significant medical disorder, contraindications to dopamine antagonists/partial agonists or MRI scanning; or clinically relevant abnormal findings at the screening assessment, as determined by the principal investigator.

### Study design
This was a single-centre, randomised, double-blind, placebo-controlled, crossover study. Two independent groups of healthy volunteers received either amisulpride and placebo (arm 1) or aripiprazole and placebo (arm 2) for 7 days each. Within each arm, the order of administration was randomised and counter-balanced to ensure approximately equal numbers received active drug and placebo first. Amisulpride doses were titrated up to 400 mg/day (day 1: 200 mg, day 2: 300 mg, days 3-7: 400 mg). Aripiprazole doses were titrated up to 10 mg/day (day 1: 5 mg, day 2: 5 mg, days 3-7: 10 mg).

Volunteers were evaluated at a screening appointment prior to randomisation. After the screening visit, eligible subjects were randomised to treatment order (amisulpride or placebo first in arm 1, aripiprazole or placebo first in arm 2). These subjects subsequently returned for the baseline assessment, following which the first dose of study medication was administered at the research facility, and the remaining 6 days of medications were dispensed to be taken at home. After completing the first treatment period, subjects returned for outcome and safety assessment after seven days, before entering a washout period of at least five half-lives of the drug and its active metabolites (minimum 10 days for amisulpride, minimum 28 days for aripiprazole). After the washout period, subjects returned to the research facility and commenced the other treatment condition. Compliance was assessed at the end of each treatment week with pill counts and serum drug levels. Only subjects with detectable amisulpride/aripiprazole levels following the active treatment week were included in the analysis sample.

### Outcome measures
Demographic information was self-reported. Outcome data (MRI data, negative symptoms, extrapyramidal symptoms) were assessed at baseline, following one week of amisulpride or aripiprazole and following one week of placebo. Subjects with ongoing adverse events during the washout period or following conclusion of the study were contacted to ensure their resolution. Outcome data were not collected prior to commencing the second treatment week (at the dosing visit), however during this visit all subjects were assessed for carry-over effects with a clinical history including adverse events, and physical examination including neurological examination (see supplement for further details). Plasma amisulpride or aripiprazole and de-hydroaripiprazole levels were measured following each treatment week, and aripiprazole and de-hydroaripiprazole levels were also measured following the washout period to detect and exclude slow metabolisers of aripiprazole. Detection was by selective reaction monitoring using tandem mass spectrometry. Instrument control was via a PC using the Agilent EZChrom, and Thermo Xcalibur software; data acquisition and processing was via the Thermo Xcalibur software.

Negative symptoms were assessed by trained clinicians blinded to treatment condition using the Brief Negative Symptom Scale (BNSS). The BNSS has high test-retest reliability, and has been shown to have either an underlying 5 factor structure (consisting of avolition, anhedonia, amotivation, alogia and blunted affect), or a 2 factor structure (in which avolition, anhedonia and amotivation load into the motivation and pleasure factor, and alogia and blunted affect load on the expressive deficits factor)[26–28,80]. Extrapyramidal symptoms were also assessed by blinded, trained clinicians using the Simpson-Angus Scale (SAS) and the Barnes Akathisia Rating Scale (BARS).

### Statistical analysis
Data were analysed in Matlab (version 9.13.0.2049777) and SPSS (v25), and plotted in GraphPad Prism (version 10.2.3). Analyses of each session were performed on all subjects with data available for that

task/procedure, other than subjects with undetectable plasma drug levels following the active treatment week who were excluded from all post-baseline analyses.

**Demographics and clinical outcomes.** Comparisons between demographic variables for the amisulpride and aripiprazole samples were performed using two-sided independent sample t-tests for continuous variables, and chi-squared tests for categorical variables.

Negative symptoms and extrapyramidal symptoms were analysed by fitting a linear mixed-effects model in Matlab using the function fitlme. The main predictor was the fixed effect of the treatment condition (active drug or placebo). We also included random intercepts and random slopes for each participant, and the fixed effects of baseline levels of the outcome variable of interest and treatment order (active drug or placebo first). The code was as follows: fitlme(Data, 'Response~DrugCondition + TreatmentOrder + Response_at_Baseline + (DrugCondition|subjectID)'). The main analysis was a modified intention-to-treat analysis; all eligible subjects who completed at least one post-baseline treatment condition were included, excluding subjects with undetectable antipsychotic levels following the active treatment week who were excluded from all post-baseline analyses. We conducted sensitivity analysis using a complete case analysis approach, and found that the significance of results was unchanged (see Supplementary for details). We compared amisulpride–placebo differences to the aripiprazole–placebo differences using two-sided independent sample t-tests for mean differences and Student's t-tests for the differences in regression slopes[81].

Correlations between measures were analysed using Pearson's correlations, or Spearman's rank correlations if data were not normally distributed as determined by the Shapiro–Wilk test. We used William's T to compare the magnitude of correlation coefficients from the same subjects which shared a common variable ($r_{12}$ and $r_{13}$), and the ZPF statistic to compare correlations from the same subjects which were nonoverlapping ($r_{12}$ and $r_{34}$)[30,31].

**Functional MRI analysis.** The FEAT module in FSL 6.00 was used to fit a generalised linear model. The contrasts of interest in the MID task were the subtraction contrasts of the reward anticipation period minus the neutral anticipation period (reward anticipation), and the reward outcome period minus the missed reward outcome period (reward outcome). The second level analysis was a two-sided paired t-test comparing the amisulpride or aripiprazole condition and placebo condition, using FSL's randomise tool with 5000 permutations and FWE correction for multiple comparisons using threshold-free cluster enhancement at $p < 0.05$.

Three bilateral striatal ROIs for the caudate, putamen, and nucleus accumbens were defined separately using the Harvard Oxford subcortical atlas, at a threshold of 50% probability. Mean parameter estimates were extracted and compared with two-sided paired samples t-tests. The change in activation in striatal regions was correlated with changes in symptom scores across the same time period and plasma drug levels. Multiple comparisons conducted across these three bilateral striatal ROIs were corrected for using the Benjamini-Hochberg False Discovery Rate correction, with a false discovery rate of 0.05[82].

A third-level analysis was conducted to compare the effects of amisulpride and aripiprazole on the MID task. This involved a mid-level analysis for each subject to calculate the drug–placebo difference using a fixed effects model in FSL. These mid-level analyses were entered into third-level analyses; two-sided independent sample t-tests using FSL's randomise tool with 5000 permutations and FWE correction for multiple comparisons using threshold-free cluster enhancement at $p < 0.05$ to generate models comparing the amisulpride–placebo difference to the aripiprazole–placebo difference, followed by pre-specified striatal ROI analyses of the above.

**Reporting summary**
Further information on research design is available in Nature Portfolio Reporting Summary linked to this article.

## Data availability
The conditions of the ethical approval of this study do not permit unrestricted access to the raw data. De-identified individual participant data are available for research purposes from the corresponding author (martin.osugo@kcl.ac.uk) from the publication date, subject to a data-sharing agreement, with the exception of data from a minority of subjects who did not consent to de-identified data being used to support future research. Requests will be responded to within 15 working days. The conditions of the ethical approval of the study stipulate that access to data which may allow identification of volunteers will only be permitted for research that has been independently reviewed by an ethics committee. Source data are provided with this paper.

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

## Acknowledgements

O.D.H. is funded by the UK Medical Research Council (grant no. MC_U120097115), the Maudsley Charity (grant no. 666), the Wellcome Trust (grant no. 094849/Z/10/Z), and the National Institute for Health Research (NIHR) Biomedical Research Centre at South London and Maudsley NHS Foundation Trust and King's College London. M.O. was supported by an NIHR Academic Foundation post and an NIHR Academic Clinical Fellowship and acknowledges support from the National Institute for Health Research (NIHR) Imperial Biomedical Research Centre (BRC). Infrastructure support was provided by the NIHR Imperial Biomedical Research Centre and the NIHR Imperial Clinical Research Facility. E.C.O. and G.E.C. were also supported by NIHR Academic Clinical Fellowships. P.S. has been supported by a Ph.D. studentship jointly funded by the NIHR-BRC at SLaM and the Department of Neuroimaging, King's College London. PS is currently supported by the European Union's Horizon 2020 Research and Innovation Program under grant agreement No. 964874 (REALMENT). The views expressed are those of the authors and not necessarily those of the NHS, the NIHR, or the Department of Health and Social Care.

## Author contributions

R.M.M., T.R.M., M.A.M., and O.D.H. designed and supervised the study. O.D.H. funded the study. M.O., P.S., U.Z., T.R.M. and O.D.H. carried out study management and administration. M.O., P.S., U.Z., V.F., G.E.C., T.W., E.C.O., and B.S. conducted study visits and acquired the data. M.B.W. provided technical assistance including setup for functional MRI imaging. M.B.W. and R.A.M. provided expert analysis support for functional MRI imaging. B.S. provided technical assistance including setup and expert analysis support for all MRI imaging. M.O. analysed all data except ASL data, which was analysed by P.S. M.O., M.B.W., P.S., U.Z., R.A.M., T.R.M., M.A.M. and O.D.H. interpreted the data. M.O. and O.D.H. wrote the manuscript, which all authors reviewed and edited.

## Competing interests

O.D.H. has received investigator-initiated research funding from and/or participated in advisory/speaker meetings organised by Angellini, Autifony, Biogen, Boehringer-Ingelheim, Eli Lilly, Elysium, Heptares, Global Medical Education, Invicro, Jansenn, Karuna, Lundbeck, Merck,

Neurocrine, Ontrack/ Pangea, Otsuka, Sunovion, Recordati, Roche, Rovi and Viatris/Mylan. He was previously a part-time employee of Lundbeck A/v. Neither O.D.H. nor his family have holdings/a financial stake in any pharmaceutical company. M.B.W. is an employee of Perceptive Inc., London. T.R.M. is an employee and founder of Pasithea Therapeutics. R.A.M. has received speaker/consultancy fees from Boehringer Ingelheim, Janssen, Karuna, Lundbeck, Otsuka, and Viatris, and co-directs a company that designs digital resources to support the treatment of mental ill health. Other authors have reported no biomedical financial interests or potential conflicts of interest.
