## [Transparent Peer Review file · Nature Communications]

Striatal dopamine D2/D3 receptor regulation of human reward processing and behaviour

Corresponding Author: Dr Martin Osugo

Version 0:

Reviewer comments:

Reviewer #1

(Remarks to the Author)

Osugo and colleagues investigate the effects of the D2/D3 antagonist amisulpride et the partial agonist aripiprazole on negative symptoms in healthy volunteers. They observe that amisulpride increases negative symptoms and reduced caudate activation during reward outcomes while no such effects are observed with aripiprazole. Both drugs cause motor side effects.

I believe that this is very important study. All previous challenge studies with antipsychotics in healthy volunteers have used single-dose design which allow only limited conclusions on potential clinical effects of modulating dopamine transmission. The authors design with treatment over one week provides valuable novel information that is of broad interest. The methods are sound and described in sufficient detail.

My main comments relate to the discussion which could provide more detail on findings that may be somewhat contrary to what could be expected. I will detail these points below. My other comments are minor.

Major points

(1) The authors explain in the introduction how dopaminergic transmission in the striatum is related to motivation and hedonic experience. In their results, they find a stronger effect of amisulpride on expressive symptoms than motivational symptoms. In the discussion, the authors almost treat these two negative symptoms interchangeably and seem to interpret the expressive deficits as impairment in motivation which is not consistent with the literature. While I am fully aware that the BNSS total score was the primary outcome, I think that these findings on the two negative symptoms domains deserve a broader and more differentiated discussion.

(2) In relation to the previous point the authors state in the abstract "sustained dopamine D2/D3 receptor antagonism results in impairments in motivated behaviour and hedonic experience". Given the fact the impairments concern more the expressive domain, I would suggest differentiating this statement a bit more. The point applied to the conclusions at the end of the discussion.

(3) The authors main fMRI findings concern the outcome phase in their reward task. While there some studies have suggested associations between activation during the outcome phase and negative symptoms, the evidence is overall stronger for the anticipation phase. One could therefore have expected an association between amisulpride effects on negative symptoms and on striatal activation in the anticipation rather than the outcome phase. I would suggest providing a more detailed discussion on why the outcome phase is concerned in the present study.

(4) In relation to the previous point, I have noted that the BOLD signal changes seem overall more prominent in the outcome phase than in the anticipation phase (figure in the supplementary materiel). This may increase the power of finding drug effects during this phase of the task. I was wondering if the authors consider that this may have played a role.

Minor points

(5) It would be helpful to mention the duration of treatment in the abstract.

(6) I fully understand the authors rationale that the doses for amisulpride and aripiprazole are both sufficient to block 60% of D2 receptors. However, in some dose equivalence schemes 400 mg of amisulpride is considered equivalent to 15 mg of aripiprazole. Would the authors consider that 10 mg of aripiprazole may be somewhat lower than the equivalent with some impact on the results?

(7) While there is certainly an argument to be made for antagonists blocking phasic and tonic transmission and for partial agonists blocking mainly phasic transmission, the evidence is to my knowledge not that clear. I would suggest toning down the respective statements a bit.

(8) In the discussion, the authors state that their results "provide strong impetus towards the development of drugs with novel mechanisms to treat psychotic disorders". While this is certainly true in a way, I believe that their results also raise the questions of a preferential use of partial agonists and of the dosing of antipsychotics. Amisulpride in lower doses may be effective for negative symptoms and more globally there is a tendency to target the minimum effective dose for maintenance treatment.

Reviewer #2

(Remarks to the Author)

This study employed a randomized, double-blind, placebo-controlled, crossover design to investigate the effects of sustained dopamine D2/D3 receptor modulation on reward processing and behavior in healthy humans. Two independent groups of volunteers participated: one receiving amisulpride (a D2/D3 antagonist) and placebo, the other receiving aripiprazole (a D2/D3 partial agonist) and placebo. Each participant underwent two 7-day treatment periods, separated by a washout period of at least five drug half-lives. The order of drug and placebo administration was randomized and counter-balanced. Doses were titrated over the week, with amisulpride reaching 400mg/day and aripiprazole reaching 10mg/day. Assessments were conducted at baseline and after each treatment period. These included:

1. fMRI scanning using the Monetary Incentive Delay task to measure neural responses to reward stimuli
2. Clinical scales (BNSS, SAS, BARS) to assess negative symptoms and motor effects
3. Blood tests for drug levels

The study design allowed for within-subject comparisons between drug and placebo conditions, as well as between-group comparisons of the effects of amisulpride versus aripiprazole. This approach enabled the researchers to isolate the specific effects of D2/D3 receptor antagonism and partial agonism on reward processing and behavior while controlling for individual differences and placebo effects.

I have three points where I'm not happy with this study and how it was conducted / how its conclusions were drawn:

1. My main criticism is about the study design and the conclusions drawn from the third-level analysis between aripiprazole and amisulpride:
 - 1.1. Separate placebo groups:

The use of separate placebo groups for amisulpride and aripiprazole introduces variability that can confound the comparison. Between-study comparisons of treatment effects can be biased due to differences in study populations, settings, and time periods.
 - 1.2. Violation of randomization:

Comparing drug effects across two separate studies violates the principle of randomization, which is crucial for causal inference. Not all participants were equally distributed among study arms, but were given the opportunity to choose between two groups (according to which rationale?).
 - 1.3. Potential for Type I error:

This approach might increase the risk of false positives. Comparing drug-placebo differences across studies could lead to spurious findings, as discussed by Ioannidis (2005) in PLoS Medicine regarding why most published research findings are false.
 - 1.4. Uncontrolled variables:

Factors such as participant characteristics, recruitment methods, and study environments might differ between the two studies, introducing confounds. However, it seems that the critical demographics are not different between the placebo groups.
 - 1.5. Lack of direct comparison:

Without a within-study comparison, it's impossible to directly attribute differences to the drugs rather than to study-specific factors. Maybe the effect is just driven by placebo-group differences? The authors should test the placebo difference.
 - 1.6. Replication issues:

This approach makes it difficult to replicate findings, a crucial aspect of scientific validity emphasized by Munafò et al. (2017) in Nature Human Behaviour.
- A more appropriate approach would have been:
- a) Conducting a single study with three arms (amisulpride, aripiprazole, placebo)
 - b) Using a crossover design where each participant receives all treatments

In conclusion, while the researchers' intent to compare these drugs is understandable, the methodology used introduces significant limitations that could undermine the validity of their conclusions. This highlights the importance of careful study design and the need for caution in interpreting between-study comparisons.

I would recommend to compare the placebo arms:

- a) Baseline comparison: It could help establish whether the two groups of participants had similar baseline responses, which is crucial for interpreting the drug effects across studies.
- b) Consistency check: It could serve as a check for consistency in task performance and neural activation patterns across the two studies.
- c) Assessing between-study variability: It could provide insight into the degree of variability between the two participant groups and study conditions.
- d) Power considerations: If the placebo arms are not significantly different, it might provide some justification for pooling data, potentially increasing statistical power.

2. It is not really clear why the authors ignored the anticipation phase and focused on the outcome phase in their MID paradigm. The MID paradigm used is a bit special in comparison to other studies e.g. it uses a win/hit rate of 50% while most studies use 66%. It is not known how reliable this paradigm is, a measure important for repeat-measure studies. I sum, there is a deviation from standard MID paradigms, which could affect comparability with other studies using the standard 66% hit rate.

Potential impacts:

- a) A lower hit rate might increase task difficulty and potentially frustration for participants.
- b) It could alter the reward expectancy, potentially affecting the anticipation phase of the task.
- c) The lower success rate might influence the motivational aspects of the task.

Rationale: The authors don't provide an explicit rationale for this change in the hit rate. It would be valuable to know why they chose this different approach. I suppose, that this makes construction of a outcome hit vs. miss contrast easier. A drawback is, that most studies of MID paradigms concentrated on the anticipation phase, but this non-standard paradigm makes evaluation of the anticipation phase more difficult.

3. While the study has some excellent sides, its conclusion seems a bit of an overstatement. I'm not sure whether the study is as new as the author state. a) The D2/D3 receptor signaling and reward processing has been well-established in previous studies. For instance, Caravaggio et al. (2019) in *Neuropsychopharmacology* showed that individual differences in D2/3 receptor availability in the ventral striatum predicted reward learning performance in healthy humans. b) Striatal involvement in motivational and hedonic behavior has been known for some time. A review by Berridge and Kringelbach (2015) in *Neuron* extensively discussed the role of striatal circuits in pleasure and motivation. c) Dissociation between reward and motor function pathways has been supported by previous research. For example, Haber (2016) in *Neuropsychopharmacology* reviews the distinct cortico-basal ganglia-thalamo-cortical circuits involved in reward processing versus motor control. However, the study does offer some novel contributions: d) Causal evidence in humans is sth. new as most of the previous work has been correlational or based on animal models. This study provides causal evidence in humans through pharmacological manipulation. This is highlighted by Webber et al. (2021) in their review of pharmacological studies of dopamine and reward, emphasizing the need for such causal studies in humans. e) Most previous studies used acute drug administration. This study examines the effects of sustained D2/D3 antagonism, which is more relevant to clinical use of antipsychotics. I'm not sure whether this is just an interesting detail for clinicians or whether it has some implications for basic research. Apart from that, the steady-state is not reached for aripiprazole. In conclusion, while the basic principles stated in the conclusion are indeed well-established, the study's main contribution lies in providing causal, integrative evidence in humans under conditions more closely mimicking clinical use of antipsychotics. However a more precise conclusion might have emphasized the study's unique contributions in terms of causal human evidence and the comparison between different types of D2/D3 modulators under sustained administration.

Reviewer #3

(Remarks to the Author)

In this manuscript, the authors describe a double-blind, placebo-controlled crossover design where two groups of healthy control participants either received amisulpride/placebo or aripiprazole/placebo for 7 days each. This study expands upon critical prior work that has indicated that D2/D3 signaling is important for reward processing and motivated behavior in humans. Prior studies employed acute drug administration designs; by assessing longer-term effects of these medications, the current study sheds light on important implications for people who are taking these medications daily. The manuscript is well-written, the study well-designed, and the results are made clear by the figures included. I only have minor comments.

1. The authors mention that their results demonstrate that D2/D3 receptors have a key role in regulating "real-world" behavior (Discussion, paragraph 2, sentence 2). It was not clear to me how the measures used in the study (i.e., fMRI MID task, etc.) can generalize to real-world behavior. Can the authors describe more about how their study exactly translates to real-world behavior?

2. The authors mentioned how plasma levels were measured following the washout period. Can the authors comment on whether they followed the participants during this time or after the 2nd administration to determine if motor side effects returned to normal?

3. There were no medication effects on task performance for either drug. Is this something the authors expected or didn't expect? Adding a comment to the Discussion might improve the readers' understanding of why brain changes would occur without corresponding behavioral task effects.

Version 1:

Reviewer comments:

Reviewer #1

(Remarks to the Author)

The authors have addressed all my comment in the revised version of my manuscript. The more detailed analyses on the 5 symptom domains provide a complex pattern of results and clearly points to aspects for which further research is needed. Nevertheless, I am comforted in my initial evaluation that this important paper will be a very valuable contribution to the literature.

Reviewer #2

(Remarks to the Author)

The only thing that leaves a little to be desired: There are a few points from regarding the design and randomisation. These points are difficult to address as the design cannot be changed, but I think the authors have made an effort to justify the design and consider possible risks associated with the design (e.g. controlled for type I error, added baseline and placebo comparisons, etc.).

Perhaps it could be addressed again that the allocation to the different arms was NOT randomised, but only the order of when the active substance was given and when placebo was given. But in general, I think the points regarding the design were dealt with as well as possible in retrospect.

Reviewer #3

(Remarks to the Author)

The authors have addressed all my concerns and that of the other reviewers' thoroughly. I have no further comments.

We thank the reviewers for their time in reviewing the paper and for their helpful comments, which we address below:

REVIEWER COMMENTS

Reviewer #1 (Remarks to the Author):

Osugo and colleagues investigate the effects of the D2/D3 antagonist amisulpride et the partial agonist aripiprazole on negative symptoms in healthy volunteers. They observe that amisulpride increases negative symptoms and reduced caudate activation during reward outcomes while no such effects are observed with aripiprazole. Both drugs cause motor side effects.

I believe that this is very important study. All previous challenge studies with antipsychotics in healthy volunteers have used single-dose design which allow only limited conclusions on potential clinical effects of modulating dopamine transmission. The authors design with treatment over one week provides valuable novel information that is of broad interest. The methods are sound and described in sufficient detail.

My main comments relate to the discussion which could provide more detail on findings that may be somewhat contrary to what could be expected. I will detail these points below. My other comments are minor.

Major points

(1) The authors explain in the introduction how dopaminergic transmission in the striatum is related to motivation and hedonic experience. In their results, they find a stronger effect of amisulpride on expressive symptoms than motivational symptoms. In the discussion, the authors almost treat these two negative symptoms interchangeably and seem to interpret the expressive deficits as impairment in motivation which is not consistent with the literature. While I am fully aware that the BNSS total score was the primary outcome, I think that these findings on the two negative symptoms domains deserve a broader and more differentiated discussion.

We thank the reviewer for highlighting this. We have amended the language in the paper, so that “negative symptoms” is used to refer to the BNSS total results, whilst deficits in motivation and hedonic experience are used only for the BNSS motivational subscale.

We also conducted additional analyses (see below) to provide a more granular assessment of negative symptom factors to address the reviewer’s concerns. We analysed the BNSS 5 factor model in addition to the 2-factor model already reported. Unfortunately, during this analysis it became apparent that the BNSS expressive deficits factor had been miscalculated (it had included lack of normal distress, which is not recommended as part of either factor (Kirpatrick, 2018: <https://doi.org/10.1016/j.schres.2017.11.031>). After recalculation of the factors, we found no relationship between either the expressive factor or the motivational factor and striatal response. We did find exploratory relationships with the BNSS avolition factor from the 5-factor scale, which has now been included in the manuscript. We apologise to the reviewers and to the journal for this oversight.

We have amended the main text as follows with the additional analyses, and corrections to the BNSS 2 factor results:

“Exploratory analysis of the BNSS five-factor model indicated that amisulpride increased ratings of blunted affect ($p=0.0073$) and alogia ($p=0.011$) compared to placebo, but not ratings of anhedonia ($p=0.35$), asociality ($p=0.10$), or avolition ($p=0.18$). “

.....

“We then went on to conduct further exploratory analyses of potential mechanisms associated with the symptom changes demonstrated. We found that the increase in total negative symptoms on the BNSS between the baseline and amisulpride assessments was highly correlated with the change in parkinsonian symptoms on the SAS over the same time period ($n=27$, $\rho=0.617$, $p<0.001$), but was not related to the change in symptoms of akathisia on the BARS ($n=27$, $\rho=0.227$, $p=0.25$). Amisulpride induced parkinsonian symptoms were related to both the expressive ($n=27$, $\rho=0.549$, $p=0.003$) and motivational domains of the BNSS ($n=27$, $\rho=0.439$, $p=0.022$). The effect size of the relationship with the expressive domain was greater numerically, but did not differ statistically from the correlation with motivational deficits ($n=27$, William’s $T=0.68$, $p=0.50$)³¹.

We found no relationship in exploratory correlations between the reduction in caudate reward signal and the increase in motor symptoms between the baseline and amisulpride assessments (SAS: $n = 21$, $\rho = -0.027$, $p = 0.91$; BARS: $n = 21$, $\rho = 0.337$, $p = 0.13$). Exploratory correlations between the change in the BNSS factors showed that the change in caudate reward signal was not correlated with the change in the BNSS expressive factor ($n = 21$, $\rho = -0.412$, $p = 0.062$) or the BNSS motivation and pleasure factors of the two-factor model ($n = 21$, $\rho = -0.396$, $p = 0.076$), but was related to the change in the BNSS avolition factor ($n=21$, $\rho=-0.459$, $p=0.037$) of the five-factor model (for full results see table S7)³² .”

...

“Compared with placebo, the dopamine D2/D3 partial agonist aripiprazole did not lead to negative symptoms overall (BNSS: $b = 1.24$, $t(48) = 1.69$, 95% CI = $-0.24:2.72$, $p=0.10$), or on the expressive ($p = 0.18$), motivational ($p = 0.14$), anhedonia ($p=0.10$), asociality ($p=0.99$), avolition ($p=0.23$), blunted affect ($p=0.30$) or alogia ($p=0.11$) subscales of the BNSS”

...

“We did not find evidence that, relative to placebo, amisulpride induced greater overall negative symptoms than aripiprazole (BNSS: $t(54) = -1.71$, $p = 0.094$). We also found no difference between amisulpride–placebo and aripiprazole–placebo difference on the expressive ($t(54) = -1.69$, $p = 0.097$) and motivation/pleasure subscales of the BNSS ($t(54) = -1.00$, $p = 0.32$). However, we found exploratory evidence that amisulpride led to greater blunted affect relative to placebo than aripiprazole ($t(54) = -2.09$, $p=0.041$), with no difference between the effects of the drugs relative to placebo on the other BNSS factors or on motor symptoms (SAS: $t(54) = 0.33$, $p = 0.74$; BARS: $t(54) = 0.97$, $p = 0.33$; figures 4A:4C).”

We have also added the following to the discussion:

“We found no relationship between striatal reward deficits and motor impairments induced by amisulpride, but found that negative symptoms induced by amisulpride were related to both variables. Specifically, we found in exploratory analyses of the BNSS 5 factor model (which reflects current consensus negative symptom domains¹) that blunted caudate D2/D3 striatal reward signalling may be particularly related to avolition, and that parkinsonian symptoms were related to both the expressive deficits and motivation/pleasure factors of the BNSS. Whilst the BNSS total score

was our primary outcome, this provides preliminary causative evidence to support current hypotheses on the pathophysiology of avolition, drawn from studies conducted in medicated schizophrenia patients which demonstrated relationships between blunted caudate response to reward and avolition, and studies in animals.²⁻⁴ However, we note that these exploratory analyses were not corrected for multiple comparisons; highlighting the need for further investigations to confirm them. The relationship between parkinsonian symptoms and both motivation/pleasure and expressive domains is more difficult to interpret, particularly as aripiprazole caused parkinsonian symptoms to a similar degree to amisulpride but did not induce negative symptoms. These data suggest that there may be multiple mechanisms underlying antipsychotic induced negative symptoms, which require further elucidation for example with more sensitive computerised measures of expressive deficits and motor function.”

(2) In relation to the previous point the authors state in the abstract “sustained dopamine D2/D3 receptor antagonism results in impairments in motivated behaviour and hedonic experience”. Given the fact the impairments concern more the expressive domain, I would suggest differentiating this statement a bit more. The point applied to the conclusions at the end of the discussion.

We thank the reviewer for highlighting this, which we have addressed as detailed for point 1).

(3) The authors main fMRI findings concern the outcome phase in their reward task. While there some studies have suggested associations between activation during the outcome phase and negative symptoms, the evidence is overall stronger for the anticipation phase. One could therefore have expected an association between amisulpride effects on negative symptoms and on striatal activation in the anticipation rather than the outcome phase. I would suggest providing a more detailed discussion on why the outcome phase is concerned in the present study.

We thank the reviewer for the above comments, which we address with the following paragraph added to the discussion:

“Although it has been hypothesised that dopaminergic signalling is more related to reward anticipation than reward outcome, the validity of these theorised phases of reward processing is unclear, as are the molecular mechanisms underlying them⁵. Schizophrenia has been associated with abnormalities in both reward anticipation and reward outcome⁶. At the time of study conception, single dose studies in healthy humans were equivocal, with one significant double-blind study showing reduced striatal reward response following acute D2/D3 antagonism for reward anticipation and reward outcome respectively (a further study showing reduced striatal response during reward anticipation has since been published⁷⁻⁹). We therefore lacked a clear hypothesis as to whether subchronic D2/D3 modulation would preferentially affect reward anticipation or reward outcome, and chose to investigate effects on both phases, as was intended in the design of the MID task¹⁰. However, we used a 50% win rate in contrast to the 66% win rate in the original version of the MID as a lower number of events per condition can lead to lower reliability¹¹. The 50% win rate used in our study has been extensively used in prior research, and studies using it have been shown to activate similar brain regions to the versions of the MID with a 66% win rate¹²⁻¹⁴. Nevertheless, a systematic investigation of the effect of hit-rate on signal magnitude during the reward anticipation and reward outcome phases of the MID has not been conducted. We detected more extensive striatal activation at baseline during the reward outcome phase compared to the reward anticipation phase, which may have increased our ability to find drug effects during the reward outcome phase. This may be attributable to the relatively low magnitude reward utilised in our study (£0.30), as some evidence suggests that striatal response in the anticipation phase is more sensitive to reward magnitude than

the reward outcome phase.^{12, 15} Overall, it is challenging to separate reward anticipation, reward outcome and reward learning in fMRI tasks, including the MID⁵, which may explain some of the inconsistencies in the acute dose studies discussed above. Future research should aim to test the effects of sustained D2/D3 modulation on reward processing using tasks which more clearly distinguish the putative phases of reward processing.”

(4) In relation to the previous point, I have noted that the BOLD signal changes seem overall more prominent in the outcome phase than in the anticipation phase (figure in the supplementary material). This may increase the power of finding drug effects during this phase of the task. I was wondering if the authors consider that this may have played a role.

We thank the reviewer for highlighting this, which we have addressed in the response to point 3)

Minor points

(5) It would be helpful to mention the duration of treatment in the abstract.

Thank you, we have done so.

(6) I fully understand the authors rationale that the doses for amisulpride and aripiprazole are both sufficient to block 60% of D2 receptors. However, in some does equivalence schemes 400 mg of amisulpride is considered equivalent to 15 mg of aripiprazole. Would the authors consider that 10 mg of aripiprazole may be somewhat lower than the equivalent with some impact on the result.

We thank the reviewer for this comment. We have added the following paragraph to the discussion to address this and point 8):

“Pharmacokinetic or pharmacodynamic differences between aripiprazole and amisulpride may account for the observed differences.....

Whilst the doses used in the study are considered the minimum clinically effective dose by the Maudsley Prescribing Guidelines, equivalent doses of antipsychotics are imprecise and 400mg of amisulpride is proposed to be equivalent to 15mg of aripiprazole by the same guidelines¹⁶. However, a recent meta-analysis found that aripiprazole at 10mg/day was very close to the near maximal response¹⁷. Moreover, 10mg of aripiprazole actually corresponds to greater striatal D2/D3 occupancy (approximately 80%) in comparison to 400mg of amisulpride (approximately 60%)¹⁸, supporting our interpretation that the drug differences observed are related to partial agonism by aripiprazole at D2/D3 receptors.

We also acknowledge that aripiprazole takes 14 days to reach steady state, whereas amisulpride reaches steady state in 3 days^{19, 20}. Although this is an important potential consideration, a PK/PD study of aripiprazole in healthy volunteers found that exposure (area under the curve over 24 hours) following 8 days of aripiprazole at 10mg daily was within 10% of exposure at 14 days²¹. Although this is another important potential consideration, a PK/PD study of aripiprazole in healthy volunteers found that exposure (area under the curve over 24 hours) following 8 days of aripiprazole at 10mg daily was within 10% of exposure at 14 days¹⁸. In our study, antipsychotic levels were collected as close as possible to outcome data, with the maximum interval of a few hours making it unlikely that plasma level fluctuations influenced the results, considering the 94-hour half-life of aripiprazole and its active metabolite. Additionally, the mean plasma concentration in our study corresponds to that

associated with approximately 80% striatal D2/D3 occupancy¹⁸. For these reasons, we think it is unlikely that an inadequate dose or duration of aripiprazole treatment explains the drug differences, although future studies could test this further. Finally, all previous double-blind investigations of the effects of antipsychotics on reward function in healthy volunteers have used only single doses, whereas our study more closely mimics the clinical use of antipsychotics by investigating repeated administration.

Our finding that amisulpride induced negative symptoms may seem surprising, as there is some evidence that low dose (50-300mg) amisulpride is effective as a treatment for negative symptoms in schizophrenia²². This has been hypothesised to relate to lower doses binding to presynaptic auto receptors to enhance dopamine transmission, despite a lack of clear mechanistic evidence to support this²². Whether the effects of amisulpride and aripiprazole on negative symptoms and reward functioning are dose and duration dependent is a critical factor for future studies to investigate, in order to support the development of translational models to predict the inducement of negative symptoms by antipsychotics.”

(7) While there is certainly an argument to be made for antagonists blocking phasic and tonic transmission and for partial agonists blocking mainly phasic transmission, the evidence is to my knowledge not that clear. I would suggest toning down the respective statements a bit.

We thank the reviewer for highlighting this. We have done so added the following line to the discussion:

“We interpreted the differences between aripiprazole and amisulpride as relating to preferential inhibition of phasic signalling by the partial agonist, in comparison to inhibition of both tonic and phasic dopamine signalling equally by the antagonist. We however acknowledge that this distinction has only been demonstrated in animals, and that it is unclear whether this is the case in humans”.

(8) In the discussion, the authors state that their results “provide strong impetus towards the development of drugs with novel mechanisms to treat psychotic disorders”. While this is certainly true in a way, I believe that their results also raise the questions of a preferential use of partial agonists and of the dosing of antipsychotics. Amisulpride in lower doses may be effective for negative symptoms and more globally there is a tendency to target the minimum effective dose for maintenance treatment.

Please see response to point 6)

Reviewer #2 (Remarks to the Author):

This study employed a randomized, double-blind, placebo-controlled, crossover design to investigate the effects of sustained dopamine D2/D3 receptor modulation on reward processing and behavior in healthy humans. Two independent groups of volunteers participated: one receiving amisulpride (a D2/D3 antagonist) and placebo, the other receiving aripiprazole (a D2/D3 partial agonist) and placebo. Each participant underwent two 7-day treatment periods, separated by a washout period of at least five drug half-lives. The order of drug and placebo administration was randomized and counter-balanced. Doses were titrated over the week, with amisulpride reaching 400mg/day and aripiprazole reaching 10mg/day.

Assessments were conducted at baseline and after each treatment period. These included:

1. fMRI scanning using the Monetary Incentive Delay task to measure neural responses to reward stimuli
2. Clinical scales (BNSS, SAS, BARS) to assess negative symptoms and motor effects
3. Blood tests for drug levels

The study design allowed for within-subject comparisons between drug and placebo conditions, as well as between-group comparisons of the effects of amisulpride versus aripiprazole. This approach enabled the researchers to isolate the specific effects of D2/D3 receptor antagonism and partial agonism on reward processing and behavior while controlling for individual differences and placebo effects.

I have three points where I'm not happy with this study and how it was conducted / how its conclusions were drawn:

1.

My main criticism is about the study design and the conclusions drawn from the third-level analysis between aripiprazole and amisulpride:

1.1. Separate placebo groups:

The use of separate placebo groups for amisulpride and aripiprazole introduces variability that can confound the comparison. Between-study comparisons of treatment effects can be biased due to differences in study populations, settings, and time periods.

We thank the reviewer for raising this point. Please see a detailed response in response to the related issues raised in points 1.2 and 1.5.

1.2. Violation of randomization:

Comparing drug effects across two separate studies violates the principle of randomization, which is crucial for causal inference. Not all participants were equally distributed among study arms, but were given the opportunity to choose between two groups (according to which rationale?).

We apologise to the reviewer for a lack of clarity in explaining the study design. The amisulpride and aripiprazole arms were conducted sequentially in the exact same setting, with the same procedures, recruitment methods and key study personnel. Participants did not choose between the amisulpride and aripiprazole arms and randomisation was therefore not violated. These are not two separate studies, but two arms of the same study. We have updated the main figure legend, clarified the text throughout the manuscript and supplements and added the following text to the main body of the text to clarify this:

“The amisulpride and placebo crossover study (arm 1) and aripiprazole and placebo crossover study (arm 2) were conducted sequentially at the same site, with no differences between the arms in terms of key study personnel, setting, recruitment strategy, inclusion/exclusion criteria, study design, data acquisition or data management/analysis (see methods and supplementary methods for further details)”

1.3. Potential for Type I error:

This approach might increase the risk of false positives. Comparing drug-placebo differences across studies could lead to spurious findings, as discussed by Ioannidis (2005) in PLoS Medicine regarding why most published research findings are false.

We thank the reviewer for highlighting this. We have conducted FDR correction throughout the study to control the type 1 error rate.

1.4. Uncontrolled variables:

Factors such as participant characteristics, recruitment methods, and study environments might differ between the two studies, introducing confounds. However, it seems that the critical demographics are not different between the placebo groups.

Please see response to point 1.1 which clarifies that the arms did not differ in design. As the reviewer has highlighted, arm 1 of the study (amisulpride and placebo crossover study) and arm 2 of the study (aripiprazole and placebo crossover study) were comparable in terms of key demographic variables, in addition to the methodology as now clearly explained above.

1.5. Lack of direct comparison:

Without a within-study comparison, it's impossible to directly attribute differences to the drugs rather than to study-specific factors. Maybe the effect is just driven by placebo-group differences? The authors should test the placebo difference.

We thank the reviewer for highlighting this. As suggested by the reviewer, we have conducted analyses on baseline differences and placebo differences. We have also compared the baseline to placebo differences between groups. We have included this in the main text as follows:

“To test whether these differences could be attributed to baseline differences between subjects in the two arms in the outcome variables, we conducted independent sample t-tests comparing baseline MID task performance measures, baseline striatal reward signal during reward anticipation and reward outcome, baseline motor symptoms and baseline negative symptoms between the two arms. There were no significant differences between subjects who were subsequently enrolled in arm 1 and subjects who were subsequently enrolled in arm 2 on any of these measures at baseline (all FDR corrected p-values >0.05, supplementary table S16). We conducted similar analyses to ensure that differences in placebo response did not contribute to the observed differences, and again found no significant differences between subjects in the amisulpride arm and subjects in the aripiprazole arm following the respective placebo conditions, or in the change from baseline assessment to the placebo assessment, in terms of task performance, striatal reward signal, motor function or negative symptoms ((all FDR corrected p-values >0.05, table S16).”

1.6. Replication issues:

This approach makes it difficult to replicate findings, a crucial aspect of scientific validity emphasized by Munafò et al. (2017) in Nature Human Behaviour.

We believe that our experiment is described in sufficient detail for to be independently replicated.

A more appropriate approach would have been:

- a) Conducting a single study with three arms (amisulpride, aripiprazole, placebo)
- b) Using a crossover design where each participant receives all treatments

In conclusion, while the researchers' intent to compare these drugs is understandable, the

methodology used introduces significant limitations that could undermine the validity of their conclusions. This highlights the importance of careful study design and the need for caution in interpreting between-study comparisons.

I would recommend to compare the placebo arms:

- a) Baseline comparison: It could help establish whether the two groups of participants had similar baseline responses, which is crucial for interpreting the drug effects across studies.
- b) Consistency check: It could serve as a check for consistency in task performance and neural activation patterns across the two studies.
- c) Assessing between-study variability: It could provide insight into the degree of variability between the two participant groups and study conditions.
- d) Power considerations: If the placebo arms are not significantly different, it might provide some justification for pooling data, potentially increasing statistical power.

We thank the reviewer for their considered comments. The issues about baseline comparisons and placebo differences have been addressed in the responses to points 1.2 and 1.5. It is interesting to consider the alternative study design with one placebo group for both amisulpride and aripiprazole. However, this design has a number of limitations as well, which is why we did not use it, as detailed in the section below which we have added to the discussion:

“Although we did not find any evidence of differences in baseline responses, placebo response, or in the change from baseline to placebo assessment between the amisulpride and aripiprazole arms, we acknowledge that this represents a potential confounder in interpreting differences in drug effects between arm 1 and arm 2. An alternative study design is a 3-intervention crossover (the same subjects administered placebo, amisulpride and aripiprazole). However, order effects (including carry over effects and practice effects), would be a greater issue than the design we used because subjects would perform each outcome measure 4 times. In addition, such a design would be practically challenging, with each arm requiring the 28-day washout of aripiprazole in order to maintain blinding, in addition to the likelihood of a higher dropout rate due to subject burden and adverse events.”

An increased dropout rate would lead to more missing data and therefore reduced statistical power. More generally, comparisons between different intervention/placebo pairs are widely accepted as a key component of evidence-based medicine, even when the study methodology and sample demographics differ substantially between pairs, unlike in our study.

2. It is not really clear why the authors ignored the anticipation phase and focused on the outcome phase in their MID paradigm. The MID paradigm used is a bit special in comparison to other studies e.g. it uses a win/hit rate of 50% while most studies use 66%. It is not known how reliable this paradigm is, a measure important for repeat-measure studies. In sum, there is a deviation from standard MID paradigms, which could affect comparability with other studies using the standard 66% hit rate.

Potential impacts:

- a) A lower hit rate might increase task difficulty and potentially frustration for participants.
- b) It could alter the reward expectancy, potentially affecting the anticipation phase of the task.
- c) The lower success rate might influence the motivational aspects of the task.

Rationale: The authors don't provide an explicit rationale for this change in the hit rate. It would be valuable to know why they chose this different approach. I suppose, that this makes contribution of an outcome hit vs. miss contrast easier. A drawback is, that most studies of MID paradigms

concentrated on the anticipation phase, but this non-standard paradigm makes evaluation of the anticipation phase more difficult.

We thank the reviewer for this comment, which we address with the following paragraph added to the discussion:

“Although it has been hypothesised that dopaminergic signalling is more related to reward anticipation than reward outcome, the validity of these theorised phases of reward processing is unclear, as are the molecular mechanisms underlying them⁵. Schizophrenia has been associated with abnormalities in both reward anticipation and reward outcome⁶. At the time of study conception, single dose studies in healthy humans were equivocal, with one significant double-blind study showing reduced striatal reward response following acute D2/D3 antagonism for reward anticipation and reward outcome respectively (a further study showing reduced striatal response during reward anticipation has since been published⁷⁻⁹). We therefore lacked a clear hypothesis as to whether subchronic D2/D3 modulation would preferentially affect reward anticipation or reward outcome, and chose to investigate effects on both phases, as was intended in the design of the MID task¹⁰. However, we used a 50% win rate in contrast to the 66% win rate in the original version of the MID as a lower number of events per condition can lead to lower reliability¹¹. The 50% win rate used in our study has been extensively used in prior research, and studies using it have been shown to activate similar brain regions to the versions of the MID with a 66% win rate¹². Nevertheless, a systematic investigation of the effect of hit-rate on signal magnitude during the reward anticipation and reward outcome phases of the MID has not been conducted. We detected more extensive striatal activation at baseline during the reward outcome phase compared to the reward anticipation phase, which may have increased our ability to find drug effects during the reward outcome phase. This may be attributable to the relatively low magnitude reward utilised in our study (£0.30), as some evidence suggests that striatal response in the anticipation phase is more sensitive to reward magnitude than the reward outcome phase.^{12, 15} Overall, it is challenging to separate reward anticipation, reward outcome and reward learning in fMRI tasks, including the MID⁵, which may explain some of the inconsistencies in the acute dose studies discussed above. Future research should aim to test the effects of sustained D2/D3 modulation on reward processing using tasks which more clearly distinguish the putative phases of reward processing.”

3. While the study has some excellent sides, it's conclusion seems a bit of an overstatement. I'm not sure whether the study is as new as the author state. a) The D2/D3 receptor signaling and reward processing has been well-established in previous studies. For instance, Caravaggio et al. (2019) in *Neuropsychopharmacology* showed that individual differences in D2/3 receptor availability in the ventral striatum predicted reward learning performance in healthy humans. b) Striatal involvement in motivational and hedonic behavior has been known for some time. A review by Berridge and Kringelbach (2015) in *Neuron* extensively discussed the role of striatal circuits in pleasure and motivation. c) Dissociation between reward and motor function pathways has been supported by previous research. For example, Haber (2016) in *Neuropsychopharmacology* reviews the distinct cortico-basal ganglia-thalamo-cortical circuits involved in reward processing versus motor control. However, the study does offer some novel contributions: d) Causal evidence in humans is sth. new as most of the previous work has been correlational or based on animal models. This study provides causal evidence in humans through pharmacological manipulation. This is highlighted by Webber et al. (2021) in their review of pharmacological studies of dopamine and reward, emphasizing the need for such causal studies in humans. e) Most previous studies used acute drug administration. This study examines the effects of sustained D2/D3 antagonism, which is more

relevant to clinical use of antipsychotics. I'm not sure whether this is just an interesting detail for clinicians or whether it has some implications for basic research. Apart from that, the steady-state is not reached for aripiprazole. In conclusion, while the basic principles stated in the conclusion are indeed well-established, the study's main contribution lies in providing causal, integrative evidence in humans under conditions more closely mimicking clinical use of antipsychotics. However a more precise conclusion might have emphasized the study's unique contributions in terms of causal human evidence and the comparison between different types of D2/D3 modulators under sustained administration.

We thank the reviewer for these suggestions for the discussion. The opening line of the discussion already emphasises the unique value of the study in terms of providing causal evidence, as follows:

“Although it has long been thought that striatal signalling to reward stimuli plays a role in motivated behaviour and hedonic responses, there has been scant causal evidence for this link in humans. Here we show, for the first time, that sustained dopamine D2/D3 receptor antagonism results in blunted striatal responses to rewards”

We have referenced the study by Caravaggio as helpfully suggested by the reviewer, having already cited the review of Webber in the discussion. We have added the following statement according to the reviewer's suggestion:

“Finally all previous double-blind investigations of the effects of antipsychotics on reward function in healthy volunteers have used only single doses, whereas our study more closely mimics the clinical use of antipsychotics by investigating repeated administration”

We have discussed the limitation of aripiprazole not being at steady state as follows:

“We also acknowledge that aripiprazole takes 14 days to reach steady state, whereas amisulpride reaches steady state in 3 days^{19,20}. Although this is an important potential consideration, a PK/PD study of aripiprazole in healthy volunteers found that exposure (area under the curve over 24 hours) following 8 days of aripiprazole at 10mg daily was within 10% of exposure at 14 days²¹. Although this is another important potential consideration, a PK/PD study of aripiprazole in healthy volunteers found that exposure (area under the curve over 24 hours) following 8 days of aripiprazole at 10mg daily was within 10% of exposure at 14 days¹⁸. In our study, antipsychotic levels were collected as close as possible to outcome data, with the maximum interval of a few hours making it unlikely that plasma level fluctuations influenced the results, considering the 94-hour half-life of aripiprazole and its active metabolite. Additionally, the mean plasma concentration in our study corresponds to that associated with approximately 80% striatal D2/D3 occupancy¹⁸. For these reasons, we think it is unlikely that an inadequate dose or duration of aripiprazole treatment explains the drug differences, although future studies could test this further.”

Reviewer #3 (Remarks to the Author):

In this manuscript, the authors describe a double-blind, placebo-controlled crossover design where two groups of healthy control participants either received amisulpride/placebo or aripiprazole/placebo for 7 days each. This study expands upon critical prior work that has indicated that D2/D3 signaling is important for reward processing and motivated behavior in humans. Prior studies employed acute drug administration designs; by assessing longer-term effects of these medications, the current study sheds light on important implications for people who are taking these medications daily. The manuscript is well-written, the study well-designed, and the results are made

clear by the figures included. I only have minor comments.

1. The authors mention that their results demonstrate that D2/D3 receptors have a key role in regulating “real-world” behavior (Discussion, paragraph 2, sentence 2). It was not clear to me how the measures used in the study (i.e., fMRI MID task, etc.) can generalize to real-world behavior. Can the authors describe more about how their study exactly translates to real-world behavior?

We thank the reviewer for highlighting this. We were referring to our measurement of negative symptoms as a measure of real-world behaviour, but we have amended the language as follows and apologise for the lack of clarity:

“Our work demonstrates that D2/D3 receptors have a key role in regulating striatal responses to reward stimuli and to *reward related* behaviour.”

2. The authors mentioned how plasma levels were measured following the washout period. Can the authors comment on whether they followed the participants during this time or after the 2nd administration to determine if motor side effects returned to normal?

We thank the reviewer for highlighting this. We have added the following sentence to the methods.

“Subjects with ongoing adverse events during the washout period or following conclusion of the study were contacted to ensure their resolution. Outcome data were not collected prior to commencing the second treatment week (at the dosing visit), however during this visit all subjects were assessed for carryover effects with a clinical history including adverse events, and physical examination including neurological examination (see supplement for further details).”

3. There were no medication effects on task performance for either drug. Is this something the authors expected or didn’t expect? Adding a comment to the Discussion might improve the readers’ understanding of why brain changes would occur without corresponding behavioral task effects.

We thank the reviewer for highlighting this. We have added the following sentence to the discussion.

“We also did not find altered task performance following amisulpride or aripiprazole, as expected and in line with a previous study using the MID following a single dose of risperidone in healthy humans⁹. This further strengthens our conclusion that the reduced BOLD activation was due to reduced striatal activation to reward, and not to the confounder of impaired performance, which is also associated with alterations in BOLD amplitude²³,”

1. Kirkpatrick B, Fenton WS, Carpenter WT, Jr., Marder SR. The NIMH-MATRICES consensus statement on negative symptoms. *Schizophr Bull.* Apr 2006;32(2):214-9. doi:10.1093/schbul/sbj053
2. Mucci A, Dima D, Soricelli A, et al. Is avolition in schizophrenia associated with a deficit of dorsal caudate activity? A functional magnetic resonance imaging study during reward anticipation and feedback. *Psychological Medicine.* 2015;45(8):1765-1778. doi:10.1017/S0033291714002943
3. Stepien M, Manoliu A, Kubli R, et al. Investigating the association of ventral and dorsal striatal dysfunction during reward anticipation with negative symptoms in patients with schizophrenia and healthy individuals. *PLoS One.* 2018;13(6):e0198215. doi:10.1371/journal.pone.0198215
4. Strauss GP, Bartolomeo LA, Harvey PD. Avolition as the core negative symptom in schizophrenia: relevance to pharmacological treatment development. *npj Schizophrenia.* 2021/02/26 2021;7(1):16. doi:10.1038/s41537-021-00145-4
5. Webber HE, Lopez-Gamundi P, Stamatovich SN, de Wit H, Wardle MC. Using pharmacological manipulations to study the role of dopamine in human reward functioning: A review of studies in

healthy adults. *Neuroscience & Biobehavioral Reviews*. 2021/01/01/ 2021;120:123-158.

doi:<https://doi.org/10.1016/j.neubiorev.2020.11.004>

6. Zeng J, Yan J, Cao H, et al. Neural substrates of reward anticipation and outcome in schizophrenia: a meta-analysis of fMRI findings in the monetary incentive delay task. *Translational Psychiatry*. 2022/10/16 2022;12(1):448. doi:10.1038/s41398-022-02201-8
7. Abler B, Erk S, Walter H. Human reward system activation is modulated by a single dose of olanzapine in healthy subjects in an event-related, double-blind, placebo-controlled fMRI study. *Psychopharmacology (Berl)*. Apr 2007;191(3):823-33. doi:10.1007/s00213-006-0690-y
8. McCabe C, Huber A, Harmer CJ, Cowen PJ. The D2 antagonist sulpiride modulates the neural processing of both rewarding and aversive stimuli in healthy volunteers. *Psychopharmacology (Berl)*. 2011;217(2):271-278. doi:10.1007/s00213-011-2278-4
9. Hawkins PCT, Zelaya FO, O'Daly O, et al. The effect of risperidone on reward-related brain activity is robust to drug-induced vascular changes. *Hum Brain Mapp*. 2021;42(9):2766-2777. doi:10.1002/hbm.25400
10. Knutson B, Fong GW, Adams CM, Varner JL, Hommer D. Dissociation of reward anticipation and outcome with event-related fMRI. *NeuroReport*. 2001;12(17):3683-3687.
11. Turner BO, Miller MB. Number of events and reliability in fMRI. *Cogn Affect Behav Neurosci*. Sep 2013;13(3):615-26. doi:10.3758/s13415-013-0178-2
12. Chen Y, Chaudhary S, Li C-SR. Shared and distinct neural activity during anticipation and outcome of win and loss: A meta-analysis of the monetary incentive delay task. *NeuroImage*. 2022/12/01/ 2022;264:119764. doi:<https://doi.org/10.1016/j.neuroimage.2022.119764>
13. Skumlien M, Freeman TP, Hall D, et al. The Effects of Acute Cannabis With and Without Cannabidiol on Neural Reward Anticipation in Adults and Adolescents. *Biological Psychiatry: Cognitive Neuroscience and Neuroimaging*. 2023/02/01/ 2023;8(2):219-229. doi:<https://doi.org/10.1016/j.bpsc.2022.10.004>
14. Skumlien M, Mokrysz C, Freeman TP, et al. Neural responses to reward anticipation and feedback in adult and adolescent cannabis users and controls. *Neuropsychopharmacology*. 2022/10/01 2022;47(11):1976-1983. doi:10.1038/s41386-022-01316-2
15. Dhingra I, Zhang S, Zhornitsky S, et al. The effects of age on reward magnitude processing in the monetary incentive delay task. *NeuroImage*. 2020/02/15/ 2020;207:116368. doi:<https://doi.org/10.1016/j.neuroimage.2019.116368>
16. Taylor DM, Barnes TRE, Young AH. *The Maudsley Prescribing Guidelines in Psychiatry*. Wiley; 2021.
17. Leucht S, Crippa A, Sifakis S, Patel MX, Orsini N, Davis JM. Dose-Response Meta-Analysis of Antipsychotic Drugs for Acute Schizophrenia. *American Journal of Psychiatry*. 2020/04/01 2019;177(4):342-353. doi:10.1176/appi.ajp.2019.19010034
18. Hart XM, Gründer G, Ansermot N, et al. Optimisation of pharmacotherapy in psychiatry through therapeutic drug monitoring, molecular brain imaging and pharmacogenetic tests: focus on antipsychotics. *World J Biol Psychiatry*. Jun 24 2024:1-123. doi:10.1080/15622975.2024.2366235
19. Winans E. Aripiprazole. *Am J Health Syst Pharm*. Dec 1 2003;60(23):2437-45. doi:10.1093/ajhp/60.23.2437
20. Sparshatt A, Taylor D, Patel MX, Kapur S. Amisulpride - dose, plasma concentration, occupancy and response: implications for therapeutic drug monitoring. *Acta Psychiatr Scand*. Dec 2009;120(6):416-28. doi:10.1111/j.1600-0447.2009.01429.x
21. Mallikaarjun S, Salazar DE, Bramer SL. Pharmacokinetics, Tolerability, and Safety of Aripiprazole following Multiple Oral Dosing in Normal Healthy Volunteers. *The Journal of Clinical Pharmacology*. 2004;44(2):179-187. doi:<https://doi.org/10.1177/0091270003261901>
22. Krause M, Zhu Y, Huhn M, et al. Antipsychotic drugs for patients with schizophrenia and predominant or prominent negative symptoms: a systematic review and meta-analysis. *European Archives of Psychiatry and Clinical Neuroscience*. 2018/10/01 2018;268(7):625-639. doi:10.1007/s00406-018-0869-3

23. Akhrif A, Geiger MJ, Romanos M, Domschke K, Neufang S. Task performance changes the amplitude and timing of the BOLD signal. *Translational Neuroscience*. 2017;8(1):182-190. doi:doi:10.1515/tnsci-2017-0025

Response to reviewers:

We thank the reviewers for their comments.

The only outstanding comment is below:

“Perhaps it could be addressed again that the allocation to the different arms was NOT randomised, but only the order of when the active substance was given and when placebo was given. But in general, I think the points regarding the design were dealt with as well as possible in retrospect.”

We believe that this point is already clear throughout the paper, as below

From Results:

“Two independent healthy volunteer cohorts received either amisulpride and placebo, or aripiprazole and placebo for seven days each in a randomised, double-blind, placebo-controlled, crossover design (figure 1). The amisulpride and placebo crossover study (arm 1) and aripiprazole and placebo crossover study (arm 2) were conducted sequentially at the same site”

From the caption to figure 1:

“The arms were conducted sequentially, with arm 1 completed prior to arm 2 commencing. The order of treatments within each arm (active drug or placebo first) was randomised and counter-balanced to ensure approximately equal numbers of subjects receiving drug or placebo first”

From methods:

“This was a single-centre, randomised, double-blind, placebo-controlled, crossover study. Two independent groups of healthy volunteers received either amisulpride and placebo (arm 1) or aripiprazole and placebo (arm 2) for seven days each. Within each arm, the order of administration was randomised and counter-balanced to ensure approximately equal numbers received active drug and placebo first.”

AND

“. After the screening visit, eligible subjects were randomised to treatment order (amisulpride or placebo first in arm 1, aripiprazole or placebo first in arm 2).”

From supplementary methods:

“Upon enrolment, subjects were randomised to treatment order (amisulpride or placebo first in arm 1, aripiprazole or placebo first in arm 2)”